# Space Group Equivariant Crystal Diffusion

**Rees Chang**[1][*], **Angela Pak**[1], **Alex Guerra**[2], **Ni Zhan**[2], **Nick Richardson**[2],
**Elif Ertekin**[3], **Ryan P. Adams**[2]

[1]Department of Materials Science & Engineering, University of Illinois at Urbana-Champaign
[2]Department of Computer Science, Princeton University
[3]Materials Research Laboratory, University of Illinois at Urbana-Champaign

## Abstract

Accelerating inverse design of crystalline materials with generative models has significant implications for a range of technologies. Unlike other atomic systems, 3D crystals are invariant to discrete groups of isometries called the space groups. Crucially, these space group symmetries are known to heavily influence materials properties. We propose SGEquiDiff, a crystal generative model which naturally handles space group constraints with space group invariant likelihoods. SGEquiDiff consists of an SE(3)-invariant, telescoping discrete sampler of crystal lattices; permutation-invariant, transformer-based autoregressive sampling of Wyckoff positions, elements, and numbers of symmetrically unique atoms; and space group equivariant diffusion of atomic coordinates. We show that space group equivariant vector fields automatically live in the tangent spaces of the Wyckoff positions. SGEquiDiff achieves state-of-the-art performance on standard benchmark datasets as assessed by quantitative proxy metrics and quantum mechanical calculations. Our code is available at `https://github.com/rees-c/sgequidiff`.

## 1 Introduction

Crystals comprise critical technologies like batteries [66], topological materials [90], electronic devices [96], photovoltaics [28], and more. Materials scientists have catalogued $\mathcal{O}(10^5)$ crystals experimentally [7] and $\mathcal{O}(10^6)$ *in silico* with density functional theory (DFT) simulation [15, 39, 79]. In contrast, the number of stable crystalline materials with five elements or less is estimated to exceed $10^{13}$, and even higher-order compositions are common in real materials [16]. Generative models offer a promising path to rapidly explore the vast space of crystals [100, 40, 62, 10].

Unlike molecules, crystals span the periodic table and exhibit discrete spatial symmetries according to one of 230 *space groups* [2]. Specifically, crystals have invariances to discrete translations, rotations, reflections, and sequences thereof that transform atoms into themselves or into identical atoms. The list of space group actions which map a point into itself is called a *stabilizer group*. When a set of points in $\mathbb{R}^3$ have conjugate stabilizer groups, the set is called a *Wyckoff position*. Importantly, as shown in Figure 1, Wyckoff positions can have zero volume, comprising points, lines, or planes. These zero volume sets are referred to as *special Wyckoff positions*. We give a more formal treatment of space groups and Wyckoff positions in Sec. 3.

Despite the fact that most existing crystal generative models ignore space groups and Wyckoff positions, they are critical for modeling real materials. Firstly, space groups and Wyckoff positions correlate strongly with materials properties; Neumann's principle states that all crystal properties share the same invariances as the crystal itself [65]. Thus even slightly perturbing atoms out of special Wyckoff positions will reduce the crystal's space group symmetry and can subsequently cause significant (even discontinuous) changes to its macroscopic properties [14, 11, 82, 9]. Secondly, we

---

[*]Correspondence to `reeswc2@illinois.edu`

39th Conference on Neural Information Processing Systems (NeurIPS 2025).

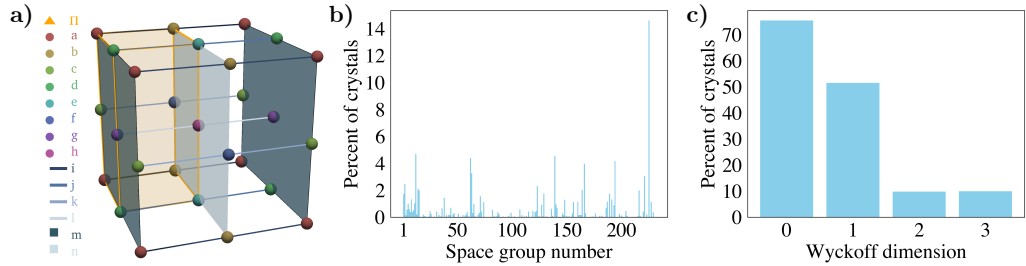

Figure 1: (a) The asymmetric unit ($\Pi$) and special Wyckoff positions labeled by letter in the conventional unit cell of space group 10. (b-c) Histograms of occupied space groups and Wyckoff dimensionalities by crystals in the MP20 [100, 39, 7] training dataset. Space groups and Wyckoff positions were determined by the `SpaceGroupAnalyzer` module in `pymatgen` [70, 91] using tolerances of 0.1 Å and 5°. These tolerances help account for the moderate convergence criteria of the Materials Project DFT relaxations.

show empirically in Figure 1 that known materials are usually invariant to high symmetry space groups with atoms in zero-volume Wyckoff positions. An intuitive explanation for this phenomenon is that atoms of the same species in a given crystal are most stable in a specific neighboring environment, and the tendency to attain this environment naturally leads to symmetry [32]. Yet, most existing crystal generative models learn continuous distributions over all three spatial dimensions of atom positions, assigning zero probability measure to placing atoms in special Wyckoff positions. One might argue that crystals from these models can be relaxed into high symmetry positions with DFT or machine learned force fields. However, besides the apparent difficulty of generating atoms sufficiently close to high symmetry positions for relaxation [107, 31, 42], such a framework assigns different model probabilities to crystals that relax into the same structure, obfuscating training and evaluation.

In this paper, we propose SGEquiDiff, which enforces space group constraints during generation and yields explicit space group invariant model likelihoods. SGEquiDiff combines SE(3)-invariant, discretized lattice sampling; permutation-invariant, transformer-based generation of occupied Wyckoff positions and elements; and space group equivariant diffusion of atomic coordinates. For the latter, we built a space group equivariant graph neural network and trained it with space group equivariant scores from a *Space Group Wrapped Normal* distribution.

The main contributions of our work are summarized as follows:

- We built a space group equivariant denoising graph neural network and regressed it against scores from a novel *Space Group Wrapped Normal* distribution.

- We prove that space group equivariant vector fields live in the tangent space of each Wyckoff position, ensuring atoms never leave their Wyckoff positions during diffusion. This result obviates the need to project atoms onto Wyckoff positions, which can lead to indeterminate probability distributions [41] or a discontinuous generation process [55, 78].

- SGEquiDiff proposes an explicit-likelihood autoregressive alternative to diffusion-based sampling of space group-constrained lattice parameters. The autoregressive sampling also enables masked in-filling. This may be useful, e.g., to generate optimal substrate materials with low lattice mismatch for epitaxial crystal growth [110, 19], commonly used for semiconductors and magnetic devices.

- We verified the efficacy of our model on standard crystal datasets with proxy metrics and rigorous DFT calculations.

## 2 Related Work

Early crystal generative models represented crystals as voxelized images [67, 34] or padded tensors of 3D coordinates [49, 75] to train variational autoencoders (VAE) [18] or generative adversarial networks [26]. Several works have enforced the SE(3) and periodic translational invariance of crystals by leveraging graph neural networks. One popular approach is to use diffusion models [83, 84, 33]

on crystal lattices, atom types, and atom positions [100, 40, 107, 57]. These models have also been extended with the flow matching framework [58], accelerating sampling [62]; the stochastic interpolants framework [1], expanding the model design space [35]; and the Bayesian Flow Networks framework [27, 97] which iteratively refines model parameters instead of samples. Other works have aimed to learn the SE(3) and periodic invariances through data augmentations [104, 31] or data canonicalization [101] with large language models (LLMs) [31, 101] or with diffusion of raw attributes [104] or of latents [42]. Recent works have used LLMs to generate noisy crystals or textual context which is passed as input to graph-based diffusion or flow matching [86, 103, 71]. We view many of these existing works as orthogonal to ours since our handling of space group symmetries can be combined with their modeling frameworks.

Two of the aforementioned works attempted to learn space group-conditioned generation without hard constraints. The graph diffusion model MatterGen [107] was fine-tuned on 14 space groups and used ground truth numbers of atoms per unit cell per space group to initialize generation. However, they could only generate stable, unique, and novel crystals in target space groups with 16% accuracy. Similarly, CrystalLLM [31] only managed 24% accuracy despite a generous `SpaceGroupAnalyzer` [91, 70] tolerance of 0.2Å.

Relevant to our model, a growing number of works have considered hard space group constraints during generation; however, to our knowledge, they either produce crystals without space group invariant likelihoods, thus assigning different likelihoods to symmetrically equivalent atoms, or rely on local structure relaxations using machine learning interatomic potentials (MLIPs) or DFT. WyCryst [108] trained a VAE to generate elements and Wyckoff position occupations, but rely on expensive DFT relaxations of atoms from uniform random locations in the Wyckoff positions. Crystal-GFN [64] considered space group constraints for the task of distribution matching under the GFlowNet framework [6] but did not address how to sample atom coordinates. DiffCSP++ [41] trained a graph-based diffusion model with masked diffusion of the unit cell lattice, continuous element diffusion with a post-hoc `argmax`, and projected diffusion of atom positions on the Wyckoff subspaces. However, DiffCSP++ is not space group equivariant to the best of our knowledge and relies on fixed templates from the training data of space group, number of atoms per unit cell, and occupied Wyckoff positions. SymmCD [55] is a non-equivariant diffusion model which leverages asymmetric units to reduce memory footprints; they use discrete diffusion of Wyckoff positions and elements and post-hoc projections of atomic coordinates onto Wyckoff positions. SymmBFN [78] recently extended SymmCD to the Bayesian Flow Networks framework [27], also requiring post-hoc projections of atom positions to the Wyckoff subspaces. CrystalFormer [10] trained an autoregressive transformer, canonicalizing crystals as a sequence of atoms ordered lexicographically by Wyckoff letter and then fractional coordinates. However, they rely on von Mises distributions which are not space group invariant. WyckoffTransformer [45] also autoregressively predicts atom types and Wyckoff positions but relies on DiffCSP++ or MLIPs to refine atoms from uniform random coordinates in the Wyckoff positions. We note that MLIPs are hindered by kinetic barriers and only preserve space group symmetry if they predict conservative forces of an invariant energy, generally requiring an expensive backward pass for inference [24, 77, 51]. WyckoffDiff [47] generates Wyckoff positions and elements with D3PM-based diffusion [3] but similarly relies on MLIPs to determine atom coordinates. Unlike these existing works, our model learns to generate complete crystals from scratch and produces space group invariant likelihoods via space group equivariant diffusion.

## 3 Preliminaries

**Space groups** Formally, a space group $G \in \mathbb{G}$, where $\mathbb{G}$ denotes the set of 230 space groups, is a group of isometries that tiles $\mathbb{R}^3$. In particular, $G$ is generated by an infinite subgroup of discrete lattice translations $T_L = \{n_1 L_1, n_2 L_2, n_3 L_3 | n_i \in \mathbb{Z}, L_i \in \mathbb{R}^3\}$ as well as a collection of other symmetry operations $g(\cdot) = \{R_g(\cdot) + v_g | R_g \in \mathrm{O}(3), v_g \in \mathbb{R}^3\} \in G$, where $R_g$ is a point group operation (rotation, reflection, or identity) and $v_g$ is a translation.

**Wyckoff positions** Given a space group and a point $x \in \mathbb{R}^3$, the *stabilizer* group $G_x \equiv \{g | gx = x\} \subset G$ (also called the site symmetry group in materials science) is the finite subgroup of $G$ that leaves $x$ invariant. A Wyckoff position is then defined as the set of points with conjugate stabilizer groups, i.e., $\{x' | \exists\, g \in G : G_{x'} = gG_x g^{-1}\}$. Conceptually, if $g$ is a point group operation, this means that all points in a Wyckoff position are invariant to the same space group operations up to a change

of basis. By convention, when $x$ is described with respect to the lattice basis $\{L_1, L_2, L_3\}$, the size of the *orbit* of $x$ in the unit cell, $|\{gx | g \in G, gx \in [0,1)^3\}|$, is called the Wyckoff *multiplicity*. Wyckoff positions whose stabilizer groups are non-trivial, i.e., include more than the identity operation, are referred to as *special Wyckoff positions* as opposed to the *general Wyckoff position* defined by the identity stabilizer group. Wyckoff positions are labeled by multiplicity and *Wyckoff letter*, where the lexicographic ordering gives the Wyckoff positions in (partial) order of increasing multiplicity.

**Unit cells**    The infinite translational periodicity of a crystal can be represented with a parallelipiped $\Gamma$ called the *unit cell*. The unit cell reduces infinite crystals by removing redundancy induced by $T_L$, the group of discrete lattice translations. In this way, crystals are represented by the tuple $M = (A, X, L)$, where $A = (a'_1, ..., a'_N) \in \mathbb{A}^N$ are the atom types, $\mathbb{A}$ is the set of all chemical elements, and $N$ is the number of atoms in the unit cell; $X = \{(x'_1, ..., x'_N) | x'_i \in \Gamma\}$ are the Cartesian atom coordinates; and $L = (L_1, L_2, L_3) \in \mathbb{R}^{3 \times 3}$ are the unit cell basis vectors. Given $M$, the infinite periodic structure can be reconstructed by applying the actions of $T_L$ as $\{(a'_i, x'_i + n_1 L_1 + n_2 L_2 + n_3 L_3)_{i=1}^N | n_j \in \mathbb{Z}\}$. Alternatively, the atom coordinates can be given in the lattice basis instead of the Cartesian basis. Coordinates in the lattice basis are commonly referred to as *fractional* coordinates. In fractional coordinates, the infinite crystal is reconstructed as $\{(a'_i, x'_i + n_1 e_1 + n_2 e_2 + n_3 e_3)_{i=1}^N | n_j \in \mathbb{Z}\}$. For the rest of this paper, we assume atom coordinates are always in fractional coordinates. While the choice of unit cell is not unique, prior crystal generative models [100, 40, 62] either canonicalize it with a minimum-volume *primitive cell* determined by the Niggli algorithm [29] or a *conventional cell* [2] which contains all the symmetries of the space group.

**Asymmetric units**    In our work, we represented crystals with a convex polytope $\Pi \in \mathbb{R}^3$, the asymmetric unit (ASU), which maximally reduces infinite crystals by removing all redundancies induced by the space group $G \supseteq T_L$. Under this formulation, we consider atoms in the ASU with fractional coordinates $X = \{(x_1, ..., x_n) | x_i \in \Pi\} \in \mathbb{R}^{n \times 3}$, atom types $A = (a_1, ..., a_n) \in \mathbb{A}^n$, and Wyckoff positions $W = \{(w_1, ..., w_n) | w_i = G_{x_i}\} \in \mathbb{W}^n$ where $n \leq N$. The infinite periodic structure of a crystal can be reconstructed by applying the actions of $G$ to $\Pi$, i.e.,

$$\{(a_i, g_{ij} x_i) \mid x_i \in \Pi, \ g_{ij} \in G/w_i, \ i \in (1, ..., n)\}.$$

By focusing on these $n$ symmetrically inequivalent atoms, we reduced our model's memory footprint and minimized the dimensionality of the generative modeling task. We canonicalized the non-unique choice of ASU using those listed in the International Tables for Crystallography [2] with additional conditions on faces, edges, and vertices from Grosse-Kunstleve et al. [30] to ensure that the ASUs are *exact*, i.e., that $\Pi$ tiles $\mathbb{R}^3$ without overlaps at the boundaries $\partial\Pi$.

**Diffusion models**    Diffusion-based generative models [33, 84] construct samples $\mathbf{x}_0 \sim p_{t=0}$ from a noisy prior $\mathbf{x}_T \sim p_{t=T}$ by evolving $\mathbf{x}_T$ through time $t$ with a learned denoising process. The training process generally involves iteratively adding noise to data samples $\mathbf{x}_0$ via the forward stochastic differential equation (SDE),

$$d\mathbf{x} = \mathbf{f}(\mathbf{x}, t)dt + \eta(t)d\mathbf{w}, \tag{1}$$

where $\mathbf{f}(\mathbf{x}, t) : \mathbb{R}^d \to \mathbb{R}^d$ is the drift coefficient, $\mathbf{w}$ is the standard Wiener process, and $\eta(t) \in \mathbb{R}$ is the diffusion coefficient. Samples are generated by running the process backwards in time from $T$ to $0$ via the *reverse* SDE,

$$d\mathbf{x} = [\mathbf{f}(\mathbf{x}, t) - \eta(t)^2 \nabla_{\mathbf{x}} \log p_t(\mathbf{x})]dt + \eta(t)d\bar{\mathbf{w}}, \tag{2}$$

where $\bar{\mathbf{w}}$ is another standard Wiener process. The generative model aims to learn the score of the marginal distribution $\nabla_{\mathbf{x}} \log p_t(\mathbf{x})$, and then simulate Eq. 2 to sample from $p_0$. In our model, we have $\mathbf{f}(\mathbf{x}, t) = 0$.

**Space group equivariant functions**    For a group of isometries $G'$, Rezende et al. [76] and Köhler et al. [52] showed that equivariant flows preserve invariant probability densities. Conversely, the gradient of a $G'$-invariant function is equivariant [44]. We are interested in using this result for the space groups. A function $f : \mathbb{R}^3 \to \mathbb{R}^3$ is space group equivariant if it satisfies $R_g f(x) = f(gx)$ for all $g \equiv (R_g, v_g) \in G$, where $R_g \in O(3)$ is a point group element and $v_g \in \mathbb{R}^3$ is a translation. *Symmetrization* via summing over group elements has become a popular method to build invariant and equivariant neural networks in the last few years [105, 74]. Recently, Mirramezani et al. [63]

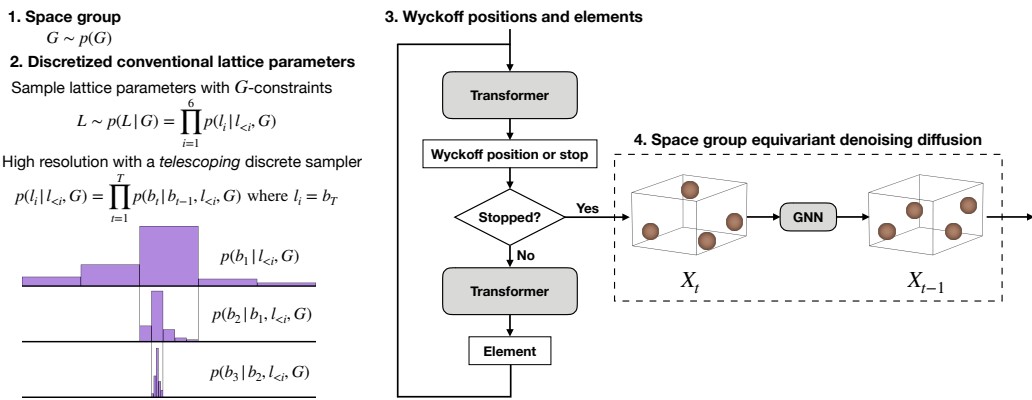

Figure 2: Illustration of our crystal generation process.

showed how symmetrization can be applied to space groups despite their countably infinite size. In particular, given the finite quotient group $G/T_L$ and a $T_L$-invariant function $\hat{f} : \mathbb{R}^3 \to \mathbb{R}^3$, the following function is space group equivariant:

$$f(x) = \frac{1}{|G/T_L|} \sum_{g \in G/T_L} R_g^{-1} \hat{f}(gx). \tag{3}$$

Besides faithfully modeling the space group symmetries of crystals, we discovered that space group equivariant vector fields conveniently live in the tangent spaces of the Wyckoff positions. We formalize this in the following theorem.

**Theorem 1**   Let $G$ be a space group, $x \in \mathbb{R}^3$ be a point residing in a Wyckoff position with stabilizer group $G_x$, and $f : \mathbb{R}^3 \to \mathbb{R}^3$ be a space group equivariant function. Then, for arbitrary constant $c \in \mathbb{R}$ and any space group element $g \in G_x$,

$$g(x + cf(x)) = x + cf(x) \implies G_x \subseteq G_{x+cf(x)}. \tag{4}$$

See proof in Sec. A.1. This result implies that a space group equivariant flow is automatically restricted to the manifold of its Wyckoff position, preventing the need for projections that lead to indeterminate probability distributions [41] or a discontinuous generation process [55, 78]. The result also implies that if, for example, a 0D Wyckoff position (a point) lies in a 1D Wyckoff position (a line), then a space group equivariant vector field cannot move an atom along the line through the point. This is problematic because we observed examples where traversing between symmetrically unique regions of a 1D Wyckoff position requires moving through a 0D Wyckoff position (e.g., in space group 192, the 1D Wyckoff position 12k is split by the 0D Wyckoff position 4c. See Fig. 3). This issue motivates SDE-based sampling since Gaussian noise (projected onto the tangent space of the Wyckoff position) can stochastically perturb atoms through 0D Wyckoff positions while maintaining that the marginal distribution over atom coordinates is still space group invariant.

## 4   Proposed method: SGEquiDiff

### 4.1   Crystal representation

Using `sympy` [61] and `PyXtal` [22], we removed redundancy induced by space group symmetry from every Wyckoff position by intersecting each one with its exact asymmetric unit. Each zero-dimensional Wyckoff position was reduced to a single point, each one-dimensional position to a set of line segments, each two-dimensional position to a set of convex polygonal ASU facets, and each three-dimensional position to the ASU interior. We will refer to these convex sets as *Wyckoff shapes*. Intersections for space group 192 are shown as an example in Figure 3. We used Wyckoff shapes for memory-efficient crystal representation and sampling as will be described in the following sections.

We decomposed each crystal into $M = (G, L, A, W, X) \in (\mathbb{G}, \mathbb{R}^{3 \times 3}, \mathbb{A}^n, \mathbb{W}^n, \mathbb{R}^{n \times 3})$, where $n$ is the number of atoms in the asymmetric unit, $G \in \mathbb{G}$ is the space group, $L \in \mathbb{R}^{3 \times 3}$ is the

conventional lattice basis, $A \in \mathbb{A}^n$ are elements, $W \in \mathbb{W}^n$ are Wyckoff positions, and $X \in \mathbb{R}^{n \times 3}$ are fractional atom coordinates. Several variables in $M$ impose hard constraints on each other. To handle these dependencies, the model sequentially samples each constrained variable after its constraining variable(s). We factorized the generation process as follows:

$$p(M) = p(G) \times p(L|G) \times p(W, A|L, G) \times p(X|W, A, L, G). \tag{5}$$

Structuring the generation process in this way rather than jointly modeling all crystal components avoids tuning per-component loss weights [100], memory-intensive mask states [104], and post-hoc projections that are intractable to marginalize over [55]. We set $p(G)$ to the empirical training distribution. Now we describe our method for the remaining conditional distributions.

## 4.2 Telescoping discrete sampling of $L$

We parameterized univariate conditionals to autoregressively sample the 3 lattice lengths $(a, b, c)$ and 3 angles between them $(\alpha, \beta, \gamma)$ for conventional unit cells, applying physical constraints to each conditional. Canonicalizing the lattice with the conventional lattice parameters makes the lattice sampling automatically SE(3) invariant. Denoting $l = (a, b, c, \alpha, \beta, \gamma)$, the model learned

$$p(L|G) = \prod_{i=1}^{6} p(l_i|l_{<i}, G), \tag{6}$$

where $p(l_i|l_{<i}, G)$ has support over positive values with finite range determined by the data. Under the space group constraints, crystal lattices can be binned into 6 crystal families, each putting unique constraints on the lattice parameters. Space groups 1 to 2 impose no constraints; 3 to 15 require $\alpha = \gamma = 90°$; 16 to 74 require $\alpha = \beta = \gamma = 90°$; 75 to 142 require $\alpha = \beta = \gamma = 90°$ and $a = b$; 143 to 194 require $a = b$, $\alpha = \beta = 90°$, and $\gamma = 120°$; and 195 to 230 require $a = b = c$ and $\alpha = \beta = \gamma = 90°$. Furthermore, the crystal lattice must have non-zero volume. To impose these constraints, the model only learns univariate conditionals for the lattice parameters unconstrained by the crystal families, leaving constrained terms in the product of Equation 6 equal to 1. We enforce positive volume by dynamically setting the support of $p(\gamma|a, b, c, \alpha, \beta, G)$ to satisfy

$$\frac{\text{Volume}}{abc} = \sqrt{1 + 2\cos\alpha\cos\beta\cos\gamma - \cos^2\alpha - \cos^2\beta - \cos^2\gamma} > 0.$$

Besides physical constraints, lattice generation requires the flexibility to learn highly peaked distributions since small perturbations to a crystal lattice can significantly alter materials properties. In $BaTiO_3$ for example, 0.03Å strain was found to increase the ferroelectric transition temperature by 500°C and the remnant polarization by 250% [13]. In contrast, the range of conventional lattice lengths in the MP20 dataset is over 100Å. To address this challenge, we chose to discretize the lattice parameters to a resolution of 0.01Å. Naively, this resolution requires a softmax over $N_l = \mathcal{O}(10^4)$ classes per lattice parameter to achieve a 100Å range. We overcame this poor scaling by *telescoping* the categorical distribution. See Figure 2 for a visual explanation. At a high level, the range of lattice parameters was first binned very coarsely, and a class $b_1$ was sampled from $p(b_1|l_{<i}, G)$. Then, the selected class $b_1$ was further coarsely binned and one of these higher resolution bins was selected from $p(b_2|b_1, l_{<i}, G)$. This process was repeated $K$ times to achieve higher levels of resolution as

$$p(l_i|l_{<i}, G) = \prod_{k=1}^{K} p(b_k|b_{k-1}, l_{<i}, G)$$

where $l_i = b_K$ and $b_0 = \emptyset$. Choosing $p(b_k|b_{k-1}, l_{<i}, G)$ to be a categorical distribution over a small number of classes $n_l \ll N_l$ achieves $(1/n_l^K) = (1/N_l)$ resolution with $\mathcal{O}(n_l K) \ll \mathcal{O}(N_l)$ memory. We used $K = 2$ and $n_l = 100$ in our experiments. We represented conditioning on $l_i$ and $b_k$ by using min-max normalization and Gaussian random Fourier features [89]. Logits for $p(b_k|b_{k-1}, l_{<i}, G)$ were produced with an MLP. Space group constraints were enforced with hard-coded masking of lattice parameters and their logits.

To better align the distributions of partially complete lattices seen during training and inference, we employed noisy teacher forcing during training. Specifically, we minimized the negative log likelihood of the next ground truth lattice parameter conditioned on noisy versions of previous ground truth lattice parameters, where noise was masked to respect space group constraints. Lattice lengths and angles were augmented with uniform random noise with 0.3Å and 5° ranges, respectively. Noising was not applied at inference time.

### 4.3 Transformer-based sampling of $W$, $A$, and $n$

Similarly to Cao et al. [10], Kazeev et al. [46], and Kelvinius et al. [47], we leveraged a transformer architecture [92] to autoregressively predict atomic Wyckoff positions $W \in \mathbb{W}^n$ and elemental species $A \in \mathbb{A}^n$ as

$$p(W, A|L, G) = \prod_{i=1}^{n} \Big[ p(w_i|w_{<i}, a_{<i}, L, G)p(a_i|w_{\leq i}, a_{<i}, L, G)p(\text{stop}|w_{\leq i}, a_{\leq i}, L, G) \Big]. \quad (7)$$

The number of atoms in the ASU $n$ is sampled implicitly via $p(\text{stop}|w_{\leq i}, a_{\leq i}, L, G)$. We used attention masking to enforce unique token orderings per crystal during training and inference. Specifically, atoms were ordered lexicographically first by Wyckoff letter and then by atomic number. The transformer input is invariant to permutations of any remaining ties. We additionally set logits of zero-dimensional Wyckoff positions which are already occupied to negative infinity to prevent sampling overlapping atoms at those locations. The transformer was fit with the cross-entropy loss. See section A.4 for details of our transformer architecture. We did not use a positional encoding since the model does not need to distinguish between different orderings of the same atoms.

To aid generalization across the conditioning variables, we initialized embeddings from physical descriptors. Element embeddings were initialized as those introduced by Xie and Grossman [99]. Similarly to the lattice sampler, min-max normalized lattice parameters were embedded with Gaussian random Fourier features. Learnable stop tokens were initialized from Gaussian noise. We represented each of the 230 space groups with 62 initial features and each of the 1731 Wyckoff positions across space groups with 231 initial features. For details and an ablation against the symmetry features used in SymmCD [55], see Sec. A.4.3.

### 4.4 Space group equivariant diffusion of $X$

We leveraged the score matching framework [84, 36] to generate atomic fractional coordinates $X$. Unlike prior works that fit to scores of the translation-invariant Wrapped Normal (WN) distribution [107, 41, 40], we learned scores of a $G$-invariant *Space Group Wrapped Normal* (SGWN) distribution:

$$p(x_t|x_0) \propto \sum_{g \in G} \exp \Big( \frac{-||x_t - gx_0||_F^2}{2\sigma_t^2} \Big). \quad (8)$$

**Theorem 2** The SGWN is $G$-invariant, i.e., $\forall g = (R_g, v_g) \in G$, $p(gx_t|x_0) = p(x_t|x_0)$. Additionally, the score is space group equivariant, i.e., $\nabla_{x_t} \log p(R_g x_t + v_g|x_0) = R_g \nabla_{x_t} \log p(x_t|x_0)$ (see Sec. A.1 for proof).

Following Jiao et al. [40], we set the noise scale $\sigma_t$ with the exponential scheduler: $\sigma_0 = 0$ and $\sigma_t = \sigma_1(\frac{\sigma_T}{\sigma_1})^{\frac{t-1}{T-1}}$ if $t > 0$. As $\sigma_t$ gets large, the SGWN approaches the uniform distribution. Noisy samples $x_t$ were created by sampling Gaussian noise $\epsilon \sim \mathcal{N}(0, I)$ and reparameterizing as $x_t = x_0 + \mathcal{P}_{x_0}(\sigma_t \epsilon)$ where $\mathcal{P}_{x_0}(\cdot)$ is an orthogonal projection to the tangent space of $x_0$'s Wyckoff shape. For the backward process, we constructed the uniform prior $p(x_T)$ on each Wyckoff position. To do so without redundantly assigning probability mass to symmetrically equivalent points, we placed uniform distributions on the relevant Wyckoff shapes. See Sec. A.3 for details. We used predictor-corrector sampling [85] to sample $x_0$.

To learn the space group equivariant scores of the SGWN distribution, we built a space group equivariant graph neural network $s_\theta$ (see Sec. A.4 for architectural details) and trained it with the following score matching loss:

$$L_X = \mathbb{E}_{x_t \sim p_t(x_t|x_0)p(x_0), t \sim \mathcal{U}(0,T)} \big[ ||s_\theta(x_t, t) - \lambda_t \nabla_{x_t} \log p(x_t|x_0)||_2^2 \big]$$

where $\lambda_t = \mathbb{E}\big[ ||\nabla_{x_t} \log p(x_t|x_0)|| \big]^{-1}$ was employed for training stability. Approximate computation of $\lambda_t$ and $\nabla_{x_t} \log p(x_t|x_0)$ is described in Sec. A.2.

To save memory when symmetrizing $s_\theta$ with Equation 3, we note that only $|G|/|G_x| \leq |G|/|T_L|$ elements in the set of points $\{gx|g \in G/T_L\}$ are unique. Thus we only predicted $\hat{f}_\theta(gx)$ on unique points and subsequently indexed into these predictions to compute the full sum over the $|G|/|T_L|$ summands in Equation 3. For points $x \in \mathbb{R}^3$ in special Wyckoff positions with large stabilizer groups $G_x$, this significantly reduced the number of forward passes through the model.

Table 1: **MP-20 dataset.** We report metrics of validity for 10,000 sampled crystals; uniqueness and novelty for 1,000 valid crystals; diversity and goodness-of-fit distributional distances for valid, unique, and novel crystals; and stability for valid, unique, and novel crystals. Stability is defined as having a DFT-predicted $E_{\text{hull}} < 0.1$ eV/atom with respect to the Materials Project v2022.8.23. $T$ represents the inference-time temperature of discrete distributions in SGEquiDiff. Unless noted otherwise, $T$=1.0. Best result is bolded, second best is underlined. Standard deviations over three training runs are reported in parentheses as errors of rightmost digits.

| | Time ↓ (s / batch) | Validity (%) ↑ | | U.N. rate (%) ↑ | | $W_\rho$ | Distribution distance ↓ | | | CMD ↓ | Diversity ↑ | | S.U.N. (%) ↑ |
| | | Structure | Composition | Template | Structure | | $W_{N_{el}}$ | $JSD_G$ | $JSD_{d_{Wyckoff}}$ | Structure | Structure | Composition | MP-2024 |
|---|---|---|---|---|---|---|---|---|---|---|---|---|---|
| CDVAE [100] | 906 | 99.99 | 85.66 | 15.5 | **98.2** | 0.6590 | 1.423 | 0.6957 | 0.4590 | 0.4821 | 0.6539 | 13.70 | 14.2 |
| DiffCSP [40] | 154 | 99.92 | 82.21 | 12.2 | 85.6 | **0.1454** | 0.4000 | 0.4638 | 0.2328 | 0.1766 | 0.9588 | 15.69 | - |
| DiffCSP++ [41] | 484 | 99.92 | 85.94 | 1.1 | 84.7 | **0.1658** | 0.5002 | **0.1608***| 0.0449*| **0.1079** | 0.9329 | 15.23 | 23.4 |
| SymmCD [55] | 139 | 88.24 | **86.76** | 9.6 | 87.7 | 0.1640 | 0.3213 | 0.1669 | 0.0344 | 0.3233 | 0.9111 | 15.62 | - |
| FlowMM† [62] | - | 96.85 | 83.19 | - | - | - | - | - | - | - | - | - | - |
| MatterGen [107] | 3,689 | **100.0** | 82.6 | 15.2 | 86.3 | **0.2059** | **0.2416** | 0.4331 | 0.2129 | 0.1338 | **0.9889** | 15.67 | 24.3 |
| SGEquiDiff | 226(12) | 99.25(28) | 86.16(32) | **18.6**(16) | 80.0(16) | **0.1938**(1049) | **0.2092**(678) | 0.1733(53) | **0.0252**(90) | 0.1718(58) | 0.9190(204) | 15.64(7) | 24.4 |
| SGEquiDiff (T=1.5) | - | 98.97(18) | 83.36(52) | **31.9**(15) | 90.1(14) | 0.3810(1440) | 0.3006(148) | 0.2183(73) | 0.0436(43) | 0.2932(92) | 0.8556(101) | 15.82(18) | - |
| SGEquiDiff (T=2.0) | - | 98.40(37) | 81.08(22) | **42.8**(31) | 95.0(2) | 0.6524(937) | 0.3685(409) | 0.2698(108) | 0.0564(61) | 0.3760(115) | 0.8439(137) | 16.28(17) | - |
| SGEquiDiff (T=3.0) | - | 97.84(53) | 78.60(35) | **57.1**(32) | 95.6(5) | 1.473(267) | 0.4962(177) | 0.3367(204) | 0.0751(41) | 0.4656(132) | 0.8177(101) | 17.16(30) | - |

\* Uses fixed templates from the training data. † Values reported by Miller et al. [62].

## 5 Experiments

**Evaluations.** We evaluated SGEquiDiff on two benchmark datasets: MP20 [100], containing 45,231 experimentally known crystals with up to 20 atoms per unit cell, and the more challenging MPTS52 dataset [4], containing 40,476 experimentally known crystals with up to 52 atoms per unit cell. Evaluations were conducted on 10,000 generated crystals. Following Xie et al. [100], we computed structural and compositional validity percentages using heuristics of interatomic distances and charge, respectively. We note that only ∼90% of crystals in MP20 pass the composition validity checker based on SMACT [16]. Of the 10,000 generated crystals, we randomly sampled 1,000 determined to be both structurally and compositionally valid. Of these 1,000 crystals, we determined how many were unique and novel with respect to the training data using pymatgen's StructureMatcher [70] with stol=0.3, angle_tol=5, and ltol=0.2 (*U.N. structures*). We also followed SymmCD and calculated the fraction of unique and novel *templates* (*U.N. templates*), where a template is defined as a space group and multiset of occupied Wyckoff positions. U.N. structures were used to compute (1) distribution distances between ground truth test and generated materials properties, including Wasserstein distances for atomic density $\rho$ and number of unique elements $N_{el}$ as well as Jensen-Shannon divergences (JSD) for space group $G$ and occupied Wyckoff dimensionalities $d_{\text{Wyckoff}}$; (2) Central Moment Discrepancy (CMD) [106] up to 50 moments between ground truth test and generated CrystalNN structural fingerprints [109]; and (3) structural and compositional diversity as measured by average pairwise $L_2$-distances between CrystalNN and Magpie [93] fingerprints, respectively. Average sampling times per batch of 500 crystals were measured on an NVIDIA A40 GPU. We list hyperparameters and training times for SGEquiDiff in Sec. A.7.

Finally, for all U.N. structures out of 1,000 random valid crystals, we conducted structure relaxations with expensive density functional theory (DFT) calculations to assess thermodynamic stability. Aligned with analyses of DFT-calculated energies for experimentally observed crystals [88] and of DFT errors relative to experiment [87, 5], stability was defined as having a predicted *energy above the hull* less than 0.1 eV/atom with respect to crystals in the Materials Project v2022.8.23 [39]. Further details on the DFT calculations are in Sec. A.8. The fraction of stable materials was multiplied by the fraction of U.N. structures to calculate the stable, unique, and novel (S.U.N.) rate. We qualify that a low energy above the hull as predicted by DFT has been found to be a necessary but insufficient condition for synthesizability [88]; developing physical theories for predictive synthesis is an active area of research in the materials science community [60, 25, 43, 12, 68, 48]. Due to the computational expense of DFT calculations, we only computed S.U.N. rates on a subset of baseline models. While prior works pre-relax generated crystals with foundational machine learning interatomic potentials (MLIPs) before conducting DFT relaxations [62, 86, 107], we opted not to use MLIPs to avoid conflating their biases [17, 53] with our evaluations of the generative models.

**Baselines.** We compared our model to several prior methods. CDVAE (4.9M parameters) [100] predicts lattices and numbers of atoms per unit cell with a VAE and then samples elements and coordinates with denoising diffusion. DiffCSP (12.3M parameters) [40] jointly diffuses the lattice with the atoms using fractional atom coordinates. MatterGen (44.6M parameters) [107] similarly diffuses the lattice, atom types, and atom positions, but leverages Cartesian atom coordinates. FlowMM [62] extends DiffCSP with the flow matching framework. DiffCSP++ (12.3M parameters) [41] extends DiffCSP by enforcing space group constraints with projected diffusion of the lattice and

Table 2: **MPTS-52 dataset.** We report metrics of validity for 10,000 sampled crystals; uniqueness and novelty for 1,000 valid crystals; and diversity and distributional distances for valid, unique, and novel crystals. Best result is bolded, second best is underlined. For DiffCSP and DiffCSP++, we used the same diffusion corrector step sizes as for MP-20, and the DiffCSP++ batch size was decreased to 96 to avoid OOMs. CDVAE failed to train due to graph construction errors from isolated atoms. Standard deviations over three training runs are reported in parentheses as errors of rightmost digits.

| | Time↓ (s / batch) | Validity (%)↑ Structure | Composition | U.N. rate (%)↑ Template | Structure | $W_\rho$ | Distribution distance↓ $W_{N_{el}}$ | $JSD_G$ | $JSD_{d_{Wyckoff}}$ | CMD↓ Structure | Diversity↑ Structure | Composition |
|---|---|---|---|---|---|---|---|---|---|---|---|---|
| DiffCSP [40] | 467 | 67.47 | 55.8 | 19.8 | 79.8 | 1.189 | 0.5006 | 0.6900 | 0.0818 | 0.4836 | **0.8621** | **16.48** |
| DiffCSP++ [41] | 1230 | **99.87** | 77.52 | 1.0 | 86.9 | 0.8244 | 0.3692 | 0.2759 | 0.1154 | **0.3812** | 0.8457 | 15.69 |
| SymmCD [55] | **210** | 87.11 | 78.18 | 14.8 | **90.2** | 1.126 | 0.3506 | **0.2730** | **0.1146** | 0.4775 | 0.7843 | 15.36 |
| SGEquiDiff | 530(40) | 97.79(37) | **79.83**(96) | **38.7**(25) | **89.8**(7) | **0.6110**(1592) | **0.1736**(428) | 0.3104(62) | 0.1300(142) | 0.4186(28) | 0.8613(292) | 16.31(11) |

atom coordinates; however, they do so without space group equivariant scores. SymmCD (60.4M parameters) [55] enforces space group constraints with non-equivariant diffusion of atoms in the asymmetric unit followed by post-hoc projections to the Wyckoff positions.

**Results.** On MP20, we found SGEquiDiff was competitive on all metrics and achieved the highest S.U.N. rate (Table 1). While MatterGen achieved a comparable S.U.N. rate, we note that it is a significantly larger model (44.6M parameters) than SGEquiDiff (5.5M parameters), sampled crystals ∼16 times slower, and yielded worse JSD metrics for space groups and Wyckoff dimensions. Compared to DiffCSP++, SGEquiDiff achieved a slightly higher S.U.N. rate and significantly higher U.N. template rate with ∼2.2 times fewer model parameters. Unsurprisingly, SGEquiDiff and other space group-constrained models (DiffCSP++, SymmCD) strongly outperformed baselines on the JSD metric for space groups and Wyckoff dimensions. We also show that by increasing the inference-time temperature of SGEquiDiff's categorical distributions over space group, Wyckoff position, element, and the stop token, the U.N. rates and composition diversity metric can be significantly improved at the expense of slightly lower validity rates and higher distributional distances to the test set. Inspecting normalized root mean square Cartesian displacements between as-generated and DFT-relaxed structures, we found all models performed similarly, with MatterGen being the best on average (Table 4). Examples of S.U.N. crystals from SGEquiDiff are shown in Fig. 4.

We also benchmarked SGEquiDiff on the more difficult MPTS52 dataset (Table 2), which most other models do not present results on. Using the same hyperparameters as used on MP20, we found that SGEquiDiff was able to scale to more atoms while maintaining a high rate of valid, unique, and novel crystals. SGEquiDiff notably achieved the highest U.N. template rate, composition validity, and Wasserstein distances for density and number of elements. Interestingly, all space group-constrained models significantly outperformed DiffCSP in the validity metrics, highlighting the advantage of enforcing nontrivial space group symmetries as an inductive bias.

**Ablations.** To isolate effects of various components of SGEquiDiff, we performed ablations on the MP-20 dataset. First, we replaced our autoregressive lattice sampler with the DDPM-based [33] sampling of E(3)-invariant lattice matrix representations proposed in DiffCSP++ and used in SymmCD (Table 6). We used the same cosine noise scheduler as DiffCSP++, jointly diffusing lattices and atom coordinates. We term this variant of SGEquiDiff as *+LDiff*. We observed that *+LDiff* improved the Wasserstein distance for number of elements (possibly because conditioning of the Wyckoff-Element transformer on lattices was removed) and CMD of structural fingerprints. However, our autoregressive lattice sampler yielded better validity, JSD of Wyckoff dimensionalities, and S.U.N. rate. We also observed that in roughly 1% of samples, the *+LDiff* model produced lattice matrices with negative determinant, resulting in a spurious inversion that does not preserve the symmetry of the 22 chiral space groups. We show examples of S.U.N. crystals from *+LDiff* in Fig. 5

We also replaced our space group and Wyckoff position features with those introduced in SymmCD [55] (Table 5). We found our featurizations yielded better structure validity, composition validity, Wasserstein distance for atomic density, structural divergence, and validation log-likelihoods of ground truth Wyckoff positions. In contrast, SymmCD features yielded more diverse samples, evidenced by higher U.N. rates and composition diversity.

Since space group symmetrization via Eq. 3 can be applied to any periodic translation invariant model, we replaced our FAENet-inspired [20] graph neural network (GNN) with the EGNN-inspired [80] model (*CSPNet*) used by other crystal generative models [40, 41, 62]. Under a fixed number of trainable parameters (2.2M), we found our GNN yielded better metrics for structural validity, composition validity, and CMD of structural fingerprints (Table 5).

**Crystal structure prediction (CSP).** Consistent with previous works [40, 41, 62], we evaluated SGEquiDiff on its ability to predict crystals from a test set using elemental composition as input. Performance was assessed by calculating match rate (MR) and root mean squared error (RMSE) of atomic fractional coordinates based on a single generated crystal per composition. RMSE was normalized by the average free length per atom. Matching was done with `pymatgen StructureMatcher` [70] using `stol=0.5`, `angle_tol=10`, and `ltol=0.3`. To adapt SGEquiDiff to this task, we combined the pretrained *+LDiff* variant of SGEquiDiff with the metric learning-based approach proposed in DiffCSP++ (*CSPML*) for selecting templates from training data and performing element substitutions to satisfy the given composition. Without CSP-specific training (unlike baselines), SGEquiDiff+LDiff achieved competitive performance (Table 3). We note SGEquiDiff could alternatively be adapted for CSP by reordering the factorization in Eq. 5 or by supplying composition as conditioning during training. While more challenging, learning instead of strictly enforcing composition may be desirable since an arbitrary composition is not guaranteed to host a stable crystal (accordingly, experimentally observed compositions can be weighted averages of multiple phases).

We highlight that the current CSP task is slightly misaligned with real-world application. Many existing models assume the number of atoms per unit cell will be known beforehand, which is not necessarily true in practice. Additionally, a single composition can host multiple stable crystal structures depending on experimental conditions. For example, iron can exist as a body-centered cubic (2 atoms per conventional cell), face-centered cubic (4 atoms per conventional cell), or at high pressures, hexagonal close-packed structure (6 atoms per conventional cell). Furthermore, materials discovery campaigns are often aimed at composition spaces with sparse training data, violating the i.i.d. assumption of the MP20 dataset splits [100].

Table 3: **Crystal structure prediction.** Leveraging the template selection and element substitution approach developed in DiffCSP++ [41], we evaluated SGEquiDiff+LDiff on the crystal structure prediction task introduced by DiffCSP [40].

| MP-20 | MR (%) ↑ | RMSE ↓ |
|---|---|---|
| CDVAE* [100] | 33.90 | 0.1045 |
| DiffCSP* [40] | 51.49 | 0.0631 |
| FlowMM* [62] | 61.39 | 0.0566 |
| DiffCSP++ (+CSPML) [41] | 70.58 | 0.0272 |
| SGEquiDiff (+LDiff,CSPML) | 69.42 | 0.0416 |

* Uses ground truth number of atoms per unit cell.

## 6 Conclusion

In this paper, we proposed SGEquiDiff to generate crystals with space group invariant likelihoods by leveraging equivariant diffusion. We showed that space group equivariant flows automatically live on the manifolds of the Wyckoff positions. Significantly, SGEquiDiff achieved state-of-the-art generation rates of stable, unique, and novel crystals as evaluated by rigorous quantum mechanical simulations. We proposed an efficient autoregressive method for sampling space group-constrained lattices. We also showed that tuning the inference-time temperature for autoregressive sampling of discrete crystal attributes provides control over the novelty of generated crystals.

**Limitations and future directions.** Due to computational constraints, we were only able to calculate S.U.N. rates on 1,000 valid crystals per model, adding unknown variance to reported metrics. SGEquiDiff's training data was pre-processed to assign space groups to crystals with nonzero tolerances over atom positions and lattice angles; it is possible that some materials were incorrectly symmetrized to higher symmetry space groups than representative of reality. SGEquiDiff also does not provide guidance for how to synthesize generated materials. Other symmetry groups relevant to crystals were ignored, including magnetic space groups [94], spin space groups [98], and layer groups [23]. Practically relevant deviations of real materials from perfect crystals were not considered, including defects, compositional disorder, surface effects, and interfacial effects. Future work might include application to a broader set of tasks such as those derived from non-scalar properties [102, 59, 95] or materials spectroscopy [56]. Furthermore, training non-equivariant models with space group invariant target distributions and/or applying inference-time space group symmetrization may enable a broader diversity of model architectures to generate high symmetry crystals.

**Broader impacts.** This work has the potential to accelerate discovery of advanced materials for energy, electronics, optics, catalysis, aerospace, and more. Possible negative impacts include development of materials that are toxic, require energy-intensive processing, lead to depletion of raw minerals, or are used for military applications.

## Acknowledgments and Disclosure of Funding

We acknowledge helpful discussions with Cindy Zhang, Barry Bradlyn, Alex M. Ganose, and Eric Toberer. This research used the Delta advanced computing and data resource (award OAC 2005572) and the Illinois Campus Cluster, operated by the Illinois Campus Cluster Program in conjunction with the National Center for Supercomputing Applications. The research was supported by the National Science Foundation under Grant Nos. DGE 21-46756 and 2118201.

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

# A  Appendix

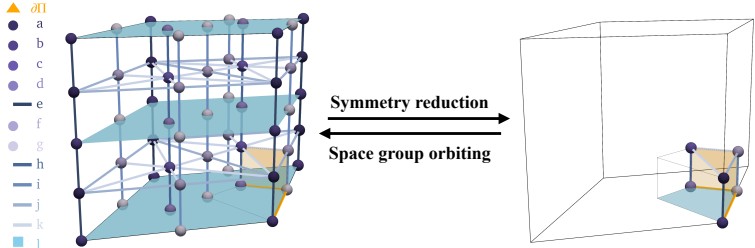

Figure 3:   Special Wyckoff positions and the asymmetric unit in the conventional unit cell of hexagonal space group 192. Closed asymmetric unit boundary edges and facets ($\partial\Pi$) are in orange.

Table 4: Normalized root mean square displacements (RMSD) in Angstroms and their standard deviations calculated with `pymatgen StructureMatcher` between as-generated and DFT-relaxed crystals. RMSD is normalized by $(V/N)^{1/3}$ for unit cell volume $V$ and number of atoms per cell $N$.

| CDVAE [100] | DiffCSP++ [41] | MatterGen [107] | SGEquiDiff | SGEquiDiff+LDiff |
|---|---|---|---|---|
| 0.0642±0.0568 | 0.0543±0.0812 | 0.0270±0.0387 | 0.0619±0.0804 | 0.0860±0.0810 |

## A.1   Proofs

**Theorem 1.**   Let $G$ be a space group, $x \in \mathbb{R}^3$ be a point residing in a Wyckoff position with stabilizer group $G_x$, and $f : \mathbb{R}^3 \to \mathbb{R}^3$ be a space group equivariant function. Then, for arbitrary constant $c \in \mathbb{R}$ and space group element $g \in G_x$,

$$g(x + cf(x)) = x + cf(x), \tag{9}$$

or equivalently, $G_x \subseteq G_{x+cf(x)}$.

*Proof.* Let $\{R_{g_x}, v_{g_x}\} = g_x \in G_x$ be a stabilizer group element where $R_{g_x} \in O(3)$ is a point group operation, $v_{g_x} \in [0, 1)^3$ is a fractional lattice translation, and the action of $g_x$ is $R_{g_x}(\cdot) + v_{g_x}$. The definition of a stabilizer group requires that

$$R_{g_x}x + v_{g_x} = x \implies v_{g_x} = 0.$$

Thus, we have

$$f(R_{g_x}x) = f(x). \tag{10}$$

The definition of space group equivariance requires that for any $R \in G$,

$$f(Rx) = Rf(x). \tag{11}$$

Combining equations 10 and 11, we see

$$R_{g_x}f(x) = f(x), \tag{12}$$

i.e., $f(x)$ is also stabilized by $G_x$. Finally, we have

$$R_{g_x}(x + cf(x)) = R_{g_x}x + cR_{g_x}f(x) \tag{13}$$
$$= x + cf(x). \tag{14}$$

**Theorem 2.**   The score of the SGWN distribution is space group equivariant, i.e., given $g = \{R, v\} \in G$,

$$\nabla_{x_t} \log p(Rx_t + v|x_0) = R\nabla_{x_t} \log p(x_t|x_0).$$

*Proof.* The score of the SGWN is given as

$$\nabla_{x_t} \log p(x_t|x_0) = \frac{\sum_{g_j=\{R_j,t_j\}\in G} \exp\left(\frac{-||x_t-(R_jx_0+t_j)||_F^2}{2\sigma_t^2}\right)(R_jx_0+t_j-x_t)}{\sigma_t^2 p(x_t|x_0)}.$$

Now, we transform the input with $g = \{R, v\}$.

$$\nabla_{x_t} \log p(Rx_t+v|x_0) = \frac{\sum_{g_j=\{R_j,t_j\}\in G} \exp\left(\frac{-||Rx_t+v-(R_jx_0+t_j)||_F^2}{2\sigma_t^2}\right)(R_jx_0+t_j-Rx_t-v)}{\sigma_t^2 p(Rx_t+v|x_0)}.$$

We will prove the equivariance of the numerator, which we denote as $\phi(Rx_t+v|x_0)$. For the denominator, it can be similarly proved that the SGWN is $G$-invariant, i.e., $p(Rx_t+v|x_0) = p(x_t|x_0)$.

$$\phi(Rx_t+v|x_0) = \sum_{\{R_j,t_j\}} \exp\left(\frac{-||Rx_t+v-(R_jx_0+t_j)||_F^2}{2\sigma_t^2}\right)(R_jx_0+t_j-Rx_t-v)$$

$$= \sum_{\{R_j,t_j\}} \exp\left(\frac{-||R(x_t-R^T(R_jx_0+t_j-v))||_F^2}{2\sigma_t^2}\right)\left[R(R^T(R_jx_0+t_j-v)-x_t)\right]$$

$$= \sum_{\{R_j,t_j\}} \exp\left(\frac{-||x_t-R^T(R_jx_0+t_j-v)||_F^2}{2\sigma_t^2}\right)\left[R(R^T(R_jx_0+t_j-v)-x_t)\right]$$

$$= \sum_{\{R_j,t_j\}} \exp\left(\frac{-||x_t-g^{-1}(R_jx_0+t_j))||_F^2}{2\sigma_t^2}\right)\left[R(g^{-1}(R_jx_0+t_j)-x_t)\right]$$

$$= \sum_{\{R_{j'},t_{j'}\}} \exp\left(\frac{-||x_t-(R_{j'}x_0+t_{j'})||_F^2}{2\sigma_t^2}\right)\left[R(R_{j'}x_0+t_{j'}-x_t)\right]$$

$$= R\phi(x_t|x_0).$$

### A.2 Implementation details

We approximated the SGWN distribution with a truncated sum over lattice translations as

$$q(x_t|x_0) \propto \sum_{t_L\in\mathbb{Z}^3\cap[-m,m]^3} \sum_{\{R_j,t_j\}\in G/T_L} \exp\left(\frac{-||x_t-(R_jx_0+t_j+t_L)||_F^2}{2\sigma_t^2}\right)$$

where $m \in \mathbb{Z}_+$. We have dropped the dependence of $q$ on $G$ and $\sigma_t$ from the notation for clarity. The space group-equivariant score of $q(x_t|x_0)$ is given as

$$\nabla_{x_t} \log q(x_t|x_0) = \nabla_{x_t} \log\left[\sum_{t_L\in\mathbb{Z}^3\cap[-m,m]^3} \sum_{\{R_j,t_j\}} \exp\left(\frac{-||x_t-(R_jx_0+t_j+t_L)||_F^2}{2\sigma_t^2}\right)\right]$$

$$= \frac{\sum_{t_L\in\mathbb{Z}^3\cap[-m,m]^3} \sum_{\{R_j,t_j\}} \exp\left(\frac{-||x_t-(R_jx_0+t_j+t_L)||_F^2}{2\sigma_t^2}\right)(R_jx_0+t_j+t_L-x_t)}{\sigma_t^2 q(x_t|x_0)}.$$

$$\tag{15}$$

For each Wyckoff position $w$, we pre-computed approximate values of $\lambda_t = \mathbb{E}_{x_t\sim p(x_t|x_0), x_0\sim\mathcal{P}_w(\text{Uniform}(\cdot))}\left[||\nabla_{x_t} \log p(x_t|x_0)||\right]^{-1}$ with Monte Carlo sampling. We denote $\mathcal{P}_w(\text{Uniform}(\cdot))$ as the uniform distribution on the Wyckoff shapes (see Sec. 4.1 and A.3) corresponding to Wyckoff position $w$. First, we sampled $s$ points $\{x_0^i\}_{i=1}^s$ from $\mathcal{P}_w(\text{Uniform}(\cdot))$. Then, for each sample $x_0^i$, we sampled Gaussian noise $\epsilon \sim \mathcal{N}(0,\sigma_t^2)$ and reparameterized as $x_t^i = x_0^i + \mathcal{P}_{x_0^i}(\epsilon)$. The approximation of $\lambda_t$ for Wyckoff position $w$ in space group $G$ was then computed as

$$\tilde{\lambda}_t = \left[\frac{1}{s}\sum_{i=1}^s \nabla_{x_t^i} \log q(x_t^i|x_0^i; G, w, \sigma_t)\right]^{-1}$$

In our experiments, we set $s = 2500$, $m = 3$, $\sigma_1 = 0.002$, $\sigma_T = 0.5$, and $T = 1000$.

### A.3 Uniform priors on Wyckoff positions

To sample from the uniform distribution in a given Wyckoff position, we sampled uniformly on the Wyckoff shapes in the ASU. The likelihoods of these uniform priors are simply given as 1 for 0D Wyckoff positions, and the reciprocal of the total length, area, and volume of the relevant Wyckoff shapes for the 1-, 2-, and 3-D Wyckoff positions, respectively. For a 0D Wyckoff position, coordinates are fully determined and 'sampling' simply returns a point. For a 1D Wyckoff position, we sampled a line segment with probability proportional to its length, and then sampled uniformly on the line segment. For a 2D Wyckoff position, we pre-computed Delaunay triangulations of the polygonal facets belonging to the Wyckoff position, sampled a triangle proportionally to its area, and then sampled from a uniform Dirichlet distribution in the triangle. For a 3D Wyckoff position, we used rejection sampling from a bounding box around the ASU.

### A.4 Architecture

#### A.4.1 Wyckoff-Element Transformer

Our encoder-decoder Transformer architecture can be summarized as follows:

$$z^0 \leftarrow \text{MLP}(e_{SG} || e_W || e_A)$$
$$z^{l+1} \leftarrow \text{EncoderLayer}(z_i^l, m_{\text{causal}})$$
$$z_W, z_A \leftarrow \text{Split}(z^{l_{\max}})$$
$$z_A \leftarrow \text{MLP}(z_A, e_W)$$
$$p_{W,\text{stop}} \leftarrow \text{Attention}(K = [e_W^{\text{all}} || e_{\text{stop}}], V = [e_W^{\text{all}} || e_{\text{stop}}], Q = z_W, \text{mask} = m_W)$$
$$p_A \leftarrow \text{Attention}(K = e_A^{\text{all}}, V = e_A^{\text{all}}, Q = z_A, \text{mask} = m_A)$$

where $e_{SG}$, $e_W$, and $e_A$ are predicted embeddings of the crystal's space group, occupied Wyckoff positions, and atomic elements, respectively; $l_{\max}$ is the number of encoder layers; $m_{\text{causal}}$ is a causal attention mask enforcing atom orderings; $e_W^{\text{all}}$, $e_{\text{stop}}$, and $e_A^{\text{all}}$ are predicted embeddings of all sampleable Wyckoff positions, the stop token, and all sampleable elements, respectively; $m_W$ and $m_A$ are attention masks enforcing lexicographic atom orderings; $p_{W,\text{stop}}$ and $p_A$ are the model probabilities over Wyckoff positions, the stop token, and elements; and $\text{EncoderLayer}(x, m)$ is a module summarized by the following:

$$x_{\text{norm}} \leftarrow \text{LN}(x)$$
$$x \leftarrow x + \text{MultiheadSelfAttention}(x_{\text{norm}}, m)$$
$$x \leftarrow \text{Dropout}(x)$$
$$x \leftarrow \text{Dropout}\big(x + \text{Linear}(\text{Dropout}(\text{MLP}(x)))\big)$$

#### A.4.2 Graph Neural Network

Our graph neural network architecture can be summarized as follows. Our GNN was a modified version of FAENet [20], replacing sum pooling with variance-preserving aggregation [81] and removing frame averaging since we trivially achieve SE(3) invariance by canonicalizing crystals with the ASU representation. We constructed fully connected atom graphs $\mathcal{G}$ wherein each atom in the conventional unit cell was connected to every other atom in the conventional unit cell by their minimum-length edge under periodic boundary conditions. If ties existed, all corresponding edges were included. At initialization, to build features which were invariant to lattice translations, we sampled frequencies $\nu_k \in \mathbb{Z}^3 \cap [-512, 512]^3 \setminus \mathbf{0}$ without replacement with probabilities drawn from

a discretized standard normal distribution. The architecture is summarized as follows:

$$t_{emb} = \text{MLP}(t)$$
$$\text{PlaneWaveEmbedding}(\cdot) \leftarrow \big[\cos(2\pi(\cdot)\nu_1)||\sin(2\pi(\cdot)\nu_1)||\cos(2\pi(\cdot)\nu_2)||\sin(2\pi(\cdot)\nu_2)||...\big]$$
$$h_i^0 \leftarrow \text{MLP}\big(e_{a_i}||\text{PlaneWaveEmbedding}(x_i)||t_{emb}\big)$$
$$e_{ij} \leftarrow \text{MLP}\big(\text{PlaneWaveEmbedding}(x_j - x_i)||\text{RBF}(d_{ij})||\text{Norm}(\mathbf{l})\big)$$
$$f_{ij}^l \leftarrow \text{MLP}\big(e_{ij}||h_i^l||h_j^l\big)$$
$$h_i^{l+1} \leftarrow h_i^l + a \cdot \text{MLP} \circ \text{GraphNorm}\left(\frac{1}{\sqrt{|\mathcal{N}_i|}}\sum_{j\in\mathcal{N}_i} h_j^l \odot f_{ij}^l\right)$$
$$h_i^{l_{\max}} \leftarrow \text{MLP}(h_i^0||...||h_i^{l_{\max}})$$
$$\hat{f}_i \leftarrow \text{MLP}(h_i^{l_{\max}})$$

where $t$ is the diffusion timestep; $x_i \in [0,1)^3$ is the fractional coordinate of atom $i$; $d_{ij}$ is the minimum-image pairwise Cartesian distance between atoms $i$ and $j$, i.e., for lattice matrix $L \in \mathbb{R}^{3\times3}$,

$$d_{ij} = \min_{n_1,n_2,n_3\in\mathbb{Z}} ||Lx_i - L(x_j + n_1 + n_2 + n_3)||;$$

$h_i^l$ is the node feature of the $i$th atom after $l$ rounds of message passing; $\mathbf{l} \in \mathbb{R}^6$ are the conventional lattice parameters; $l_{\max}$ is the number of message passing layers; $a$ is a learnable scalar initialized to zero; $e_{a_i}$ is the embedding of atom $i$'s element; and $\hat{f}_i \in \mathbb{R}^3$ is the non-symmetrized score prediction (see Eq. 3) for atom $i$. For the LDiff variant of our model, we additionally had

$$\epsilon_L \leftarrow \text{MLP}\left(\frac{1}{N}\sum_{i=1}^N h_i^{l_{\max}}\right)$$

where $\epsilon_L \in \mathbb{R}^6$ is the predicted noise for the lattice $k$-vector (see [41]).

We see that $\hat{f}_i$ is invariant to lattice translations since the model's spatial inputs are the minimum-image distances $d_{ij}$ under periodic lattice translations and translation invariant plane wave embeddings of relative and absolute atomic fractional coordinates ($x_j - x_i$ and $x_i$, respectively). We symmetrized $\hat{f}_i$ with space group equivariance according to Equation 3.

### A.4.3 Space group and Wyckoff position features

To represent a space group, we created one-hot features indicating the 6 lattice centering types, 6 crystal families, 32 crystallographic point groups, whether the space group is chiral, and whether the space group is centrosymmetric. Since this representation ignores fractional translations from glide and screw symmetries, we counted the maximum number of collisions between space group representations (16), arbitrarily indexed colliding space groups, and created additional one-hot features to avoid collisions. This yielded $6 + 6 + 32 + 1 + 1 + 16 = 62$ features. While the last 16 features are not physically motivated, they give the model the capacity to differentiate space groups and learn their relationships. These features are more abstracted than those of SymmCD [55], which used binary features indicating the 26 space group symmetry operations, 15 symmetry axes, and 7 lattice systems, totaling $7 + 26 \times 15 = 397$ features.

To featurize a Wyckoff position, we used our 62 space group features; one-hot features for the 100 unique site symmetry symbols across space groups, the 4 numbers of degrees of freedom in atom coordinates (0, 1, 2, or 3), and the 17 unique Wyckoff multiplicities; and 48 Fourier features of Wyckoff shapes' midpoints and vertices (see Sec 4.1). This yielded $62 + 100 + 4 + 17 + 48 = 231$ Wyckoff position features. To compute Fourier features $\nu \in \mathbb{R}^{48}$, we collected the midpoint and vertices of each Wyckoff shape belonging to the Wyckoff position. We used these points

$\{p_i \in \mathbb{R}^{3\times 1}\}_{i=1}^{n_w}$ to compute the following:

$$f = 2\pi \cdot \text{linspace}(\text{start} = 1, \text{end} = 32, \text{steps} = 8) \in \mathbb{R}^{1\times 8}$$

$$\nu_i = \text{flatten}(p_i f) \in \mathbb{R}^{24}$$

$$\nu = \frac{1}{n_w} \sum_i^{n_w} [\sin(\nu_i), \cos(\nu_i)] \in \mathbb{R}^{48}$$

We confirmed that this representation yielded no collisions between featurizations of different Wyckoff positions. Our representation differs from that of SymmCD, which represents each Wyckoff position by one-hot features of the 15 symmetry axes and 13 symmetry operations (totaling $15 \times 13 = 195$ features) corresponding to the Wyckoff position's stabilizer group.

## A.5 Ablation results

Table 5: **SGEquiDiff Ablations on MP-20.** We replaced our space group and Wyckoff position features with those of SymmCD [55] (*+SymmCD features*) and our denoising graph neural network with that of DiffCSP [40], DiffCSP++ [41], and FlowMM [62] (*+CSPNet*). For fair comparison in the latter experiment, we fixed the number of message passing steps (5), the number of Gaussian edge frequencies (96), and the number of trainable parameters (2.2M) by reducing CSPNet's hidden dimension from 512 to 224. Standard deviations over three training runs are reported in parentheses as errors of rightmost digits.

| | Validation Wyckoff log-likelihood ↑ | Validity (%) ↑ Structure | Composition | U.N. rate (%) ↑ Template | Structure | Distribution distance ↓ $W_\rho$ | $W_{N_{el}}$ | $\text{JSD}_{d_{\text{Wyckoff}}}$ | CMD ↓ Structure | Diversity ↑ Structure | Composition |
|---|---|---|---|---|---|---|---|---|---|---|---|
| SGEquiDiff | **-0.3206**$_{(83)}$ | **99.25**$_{(28)}$ | **86.16**$_{(32)}$ | 18.6$_{(16)}$ | 80.0$_{(16)}$ | **0.1938**$_{(1049)}$ | 0.2092$_{(678)}$ | 0.0252$_{(90)}$ | **0.1718**$_{(58)}$ | 0.9190$_{(204)}$ | 15.64$_{(7)}$ |
| +*SymmCD features* | -0.4925$_{(30)}$ | 98.55$_{(21)}$ | 82.82$_{(28)}$ | **30.0**$_{(21)}$ | **84.9**$_{(18)}$ | 0.4217$_{(1525)}$ | 0.1706$_{(279)}$ | 0.0395$_{(138)}$ | 0.3307$_{(125)}$ | 0.9078$_{(52)}$ | **16.10**$_{(30)}$ |
| +*CSPNet* | - | 99.00$_{(5)}$ | - | - | 80.6$_{(18)}$ | - | - | - | 0.1823$_{(21)}$ | 0.9252$_{(155)}$ | - |

Table 6: **LDiff ablation on MP-20.** We replaced our autoregressive lattice sampling method with joint diffusion of atom coordinates and E(3)-invariant lattice matrix representations proposed in DiffCSP++ [41] (*+LDiff*). We report metrics of validity for 10,000 sampled crystals (including the percentage of generated lattices with positive determinant, a requirement to enforce symmetry of chiral space groups); uniqueness and novelty for 1,000 valid crystals; diversity and goodness-of-fit distributional distances for valid, unique, and novel crystals; and stability for valid, unique, and novel crystals. Stability is defined as having a DFT-predicted $E_{\text{hull}} < 0.1$ eV/atom with respect to the Materials Project v2022.8.23. Best result is bolded, second best is underlined. Standard deviations over three training runs are reported in parentheses as errors of rightmost digits.

| | Time ↓ (s / batch) | $|L| > 0$ | Validity (%) ↑ Structure | Composition | U.N. rate (%) ↑ Template | Structure | $W_\rho$ | Distribution distance ↓ $W_{N_{el}}$ | $\text{JSD}_G$ | $\text{JSD}_{d_{\text{Wyckoff}}}$ | CMD ↓ Structure | Diversity ↑ Structure | Composition | S.U.N. (%) ↑ MP-2024 |
|---|---|---|---|---|---|---|---|---|---|---|---|---|---|---|
| SGEquiDiff | 226$_{(12)}$ | **100**$_{(0)}$ | **99.25**$_{(28)}$ | **86.16**$_{(32)}$ | 18.6$_{(16)}$ | 80.0$_{(16)}$ | 0.1938$_{(1049)}$ | 0.2092$_{(678)}$ | 0.1733$_{(53)}$ | **0.0252**$_{(90)}$ | 0.1718$_{(58)}$ | 0.9190$_{(204)}$ | 15.64$_{(7)}$ | **25.80** |
| +*LDiff* | 227$_{(12)}$ | 98.95$_{(49)}$ | 97.02$_{(97)}$ | 84.10$_{(63)}$ | 20.3$_{(20)}$ | 80.3$_{(10)}$ | 0.1435$_{(613)}$ | **0.1292**$_{(86)}$ | 0.1847$_{(122)}$ | 0.0579$_{(151)}$ | **0.1549**$_{(72)}$ | 0.9474$_{(131)}$ | 15.87$_{(17)}$ | 13.65$^*$ |

$^*$ For efficiency, stability calculations were conducted on 100 random valid, unique, and novel structures.

## A.6 Examples of S.U.N. crystals

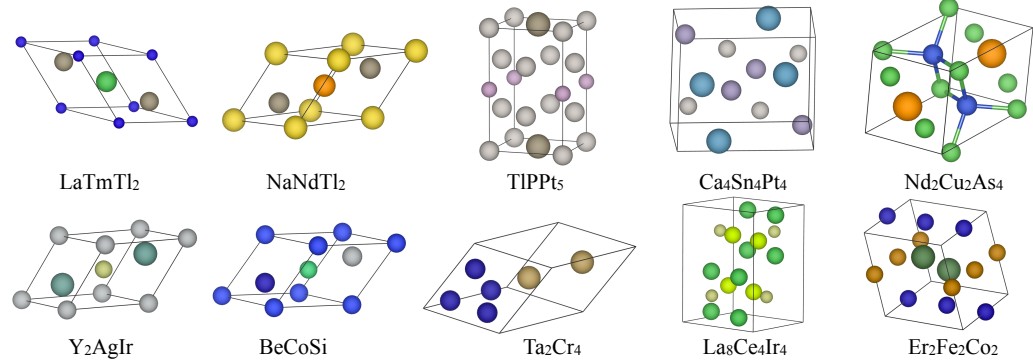

Figure 4: S.U.N. crystals generated by SGEquiDiff trained on MP20.

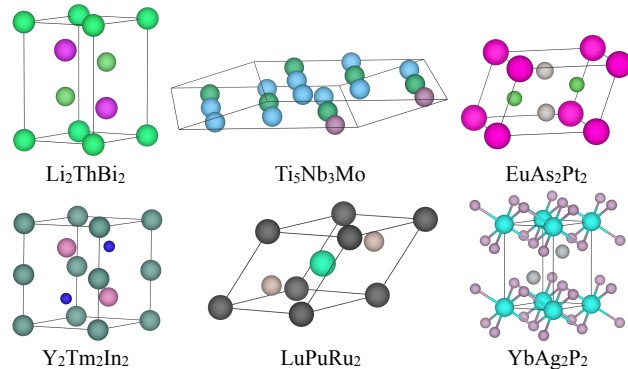

Figure 5: S.U.N. crystals generated by the *+LDiff* variant of SGEquiDiff trained on MP20.

## A.7 Training and hyperparameters

Our code was written with PyTorch [72] and PyTorch Geometric [21]. The model was trained with the Adam optimizer [50] on a single NVIDIA A40 GPU. Training SGEquiDiff took approximately 10 hours on MP20 and 12 hours on MPTS52. Data splits were the same as provided by Xie et al. [100] and Baird et al. [4]. Hyperparameters were tuned manually.

We adopted a modular architecture, simultaneously but independently training the space group sampler, lattice sampler, Wyckoff-Element transformer, and denoising graph neural network. Early stopping based on validation performance was applied to each module separately, eliminating the need for gradient balancing between modules.

| Hyperparameter | Value |
|---|---|
| Batch size | 256 |
| Number of epochs | 1000 |
| Teacher forced lattice length noise range | 0.3 |
| Teacher forced lattice angle noise range | 5.0 |
| Lattice sampler hidden dimension | 256 |
| Lattice length Fourier scale | 5.0 |
| Lattice angle Fourier scale | 1.0 |
| Lattice length bin edges Fourier scale | 2.0 |
| Lattice angle bin edges Fourier scale | 1.0 |
| Transformer dropout rate | 0.1 |
| Transformer hidden dimension | 256 |
| Transformer number of hidden layers | 4 |
| Transformer number of heads | 2 |
| SGWN lattice translations | 3 |
| SGWN Monte Carlo samples | 2500 |
| Time embedding dimension | 128 |
| Number of plane wave frequencies | 96 |
| Number of Cartesian distance radial basis functions | 96 |
| Radial basis functions cutoff distance | 10.0 |
| Edge embedder hidden dimension | 128 |
| Node embedder hidden dimension | 256 |
| Number of message passing steps | 5 |
| Space group learning rate | $1 \times 10^{-3}$ |
| Lattice learning rate | $1 \times 10^{-4}$ |
| Transformer learning rate | $1 \times 10^{-4}$ |
| GNN learning rate | $1 \times 10^{-3}$ |
| Weight decay | 0.0 |
| Learning rate scheduler | ReduceLROnPlateau |
| Scheduler factor | 0.6 |
| Scheduler patience | 30 |
| Minimum learning rate | $1 \times 10^{-5}$ |
| Gradient clipping by value | 0.5 |
| Number of parameters | 5,513,396 |

## A.8 Density Functional Theory Calculations

Total energy calculations were performed with density functional theory (DFT) using the Vienna Ab Initio Simulation Package (VASP) [54] with Projector Augmented Wave (PAW) pseudopotentials [8]. The Perdew-Burke-Ernzerhof (PBE) [73] exchange-correlation functional was used for structural relaxations. All input parameters, plane wave cutoffs, convergence criteria, and k-point densities were determined by `pymatgen` `MPRelaxSet` [69].

Relaxations which don't converge with default settings from VASP (version 6.4) and `MPRelaxSet` do not necessarily indicate issues with generated crystal structures. Specifically, we encountered and attempted to resolve three failure modes during relaxations: errors related to symmetry-finding, issues converging energy and electronic charge density within DFT's self-consistent field (SCF) loops, and issues converging atomic positions. To resolve symmetry-finding errors (reported as `SGRCON` or `INVGRP` in VASP's output files), we tightened the tolerance of the symmetry detection algorithm by setting `SYMPREC=1E-6` from the default `1E-5`. To resolve SCF convergence issues, we changed the electronic minimization algorithm to the slower but more stable `ALGO=NORMAL` from the default `ALGO=FAST`. If SCF-related issues persisted, we additionally set `AMIX=0.2` from the default `0.4`, which results in slower but more stable convergence of the charge density, and increased the number of electronic optimization steps with `NELM=200` (default `100`). Finally, to resolve issues related to convergence of atomic positions, we changed the structure optimization algorithm to the slower but more precise `IBRION=1` (default `2`). We emphasize that these changes in VASP parameters will change the relaxation trajectory but not change the minimal energy structure or loosen convergence criteria relative to `MPRelaxSet`'s default settings.

For direct comparison with Materials Project (MP) data, we applied the MP's energy correction scheme for anions and mixing GGA/GGA+$U$ calculations [37, 38]. Crystal stabilities were determined through convex hull analyses using all competing phases available in MP v2022.8.23 [39].

