# OpenReview forum: "Space Group Equivariant Crystal Diffusion"
_NeurIPS.cc/2025/Conference — NeurIPS 2025 poster_

### Official Review · Reviewer_mXGs · 2025-06-15

**Clarity:** 3
**Significance:** 2
**Originality:** 3
**Rating:** 5
**Confidence:** 4

**Summary:**

The paper proposed a novel methodology to leverage the symmetry constraints in materials to generate more accurate and symmetric crystals. In particular, the contributions include a novel autoregressive sampling of lattice parameters, transformer-based sampling of wyckoff positions and atoms, and finally, a space group equivariant diffusion of fractional coordinates. The authors benchmark their proposed model with relevant baselines in the MP20 and MPTS52 datasets.

**Questions:**

Please refer to the **Strengths And Weaknesses** section for relevant questions alongside weaknesses.

**Ethical Concerns:**

["NO or VERY MINOR ethics concerns only"]

**Final Justification:**

All my major concerns about missing information or experiments are addressed appropriately. The authors have also clarified the questions on design choices. Overall, I recommend acceptance of the paper due to its potential strong impact on the field of AI for crystal generation.

**Limitations:**

Yes. Please refer to the **Strengths And Weaknesses** section for more details.

**Quality:**

3

**Strengths And Weaknesses:**

**Strengths**:

- The paper is well-motivated, clearly structured, and effectively builds on prior work in the literature.
- The authors make thoughtful design choices that are aligned with the structure of the underlying data. Notably, the design of the denoising network, avoiding atom overlaps in the transformer-based generative model, the use of telescoping discrete sampling of lattice parameters guided by the MP20 dataset distribution, and providing a compelling rationale for avoiding uniform distribution over atom positions.
- The observation that equivariant vector fields naturally reside in the tangent spaces of Wyckoff positions is a valuable insight that could inform the future design of space group equivariant architectures.
- The proposed model achieves competitive performance on the MP20 dataset and outperforms existing methods on the more challenging MPTS52 benchmark, demonstrating its capacity to accurately generate larger, more complex materials.








**Weaknesses and Questions**:
- Line 205: The paper states "...p(G) as a simple vector of logits and trained it with the cross-entropy loss...," but it is unclear what the loss is computed between. Why not sample directly from the empirical distribution of space groups in the training dataset during generation?
- While the factorization in Equation 5 helps structure the generative process, the challenges or limitations of jointly modelling all components of $M$ are not discussed. What specific difficulties would arise in learning a joint distribution?
- The sampling of $n$ using the transformer in Section 4.3 is not described and should be detailed to ensure clarity and reproducibility.
- The model imposes a specific ordering on atoms, even though materials are fundamentally unordered sets. Would it be more scientifically appropriate to avoid this imposed ordering altogether?
- The Wyckoff position embeddings are not clearly explained. Specifically, how were the 62 features for 230 space groups and the 231-dimensional embeddings for 1731 Wyckoff positions derived? Additionally, is 1731 the total number of Wyckoff positions across all space groups?
- Using the symmetry representation from SymmCD could be a viable alternative to reduce redundancy and minimize the token space. Was any ablation performed on token space design choices that led to the current setup in Section 4.3?
- Is adding the LDiff functionally equivalent to using the invariant representation of lattice matrices introduced in DiffCSP++?
- **Questions on Table 1**:
  - What does $T$ represent in the table?
  - What is the value of $T$ in the "SGEquiDiff" row?
  - There appears to be an inverse relationship between composition validity and the unique and novel (U.N.) rate. Is there any explanation for this?
  - It is incorrectly stated that SymmCD uses fixed templates.
  - Please report the variance across multiple random seeds for SGEquiDiff (the best model per the S.U.N rate).
  - Why is the Wasserstein distance for atomic density significantly higher for SGEquiDiff compared to the baselines?
  - The table caption would benefit from being more descriptive and including relevant contextual details.
- Since S.U.N. rates are not reported for all baselines, highlighting the percentage improvement over the "next best" model is misleading. Moreover, given the acknowledged variability in S.U.N. rates (as noted in the limitations), the Lines 334–335 statement is not well-supported.
- Beyond computational efficiency, are there other advantages to using the asymmetric unit (ASU) over the conventional unit cell? Although more computationally demanding, the conventional cell may capture more structural information.


I am willing to improve the scores if my questions are adequately addressed during the rebuttal.

---

> ### Author Rebuttal · Authors · 2025-07-31
>
> Thank you for the helpful feedback. Below we aim to address your concerns. Please let us know if we can clarify anything further.
>
> > Why not sample the training distribution of space groups?
>
> We agree. For simplicity, we have changed our code to sample space groups directly from the training set, observing no changes to our results.
>
> > What difficulties would arise in learning a joint distribution?
>
> We believe there are a few difficulties. (1) Joint modeling requires tuning per-component loss weights. (2) Jointly modeling number of atoms with atom attributes requires unnatural “mask states”. (3) Space groups impose hard constraints on lattices (lines 212-217), available Wyckoff positions (lines 120-130), and atom coordinates. Sampling these constrained and constraining variables jointly will yield constraint violations with high probability. Several works try learning space group constraints instead of enforcing them, but we explain in lines 21-45 and show empirically through the $JSD_G$ and $JSD_{d_{Wyckoff}}$ metrics why this might not be a good idea.
>
> While SymmCD diffuses atomic coordinates jointly with Wyckoff positions, the model requires post-hoc projections to correct constraint violations. This complicates the model distribution (since marginalizing out projections is intractable) and possibly contributes to the lower structural validity metric (Table 1) as the model cannot recover from projections to invalid structures.
>
> > sampling $n$ using the transformer in Section 4.3 is not described
>
> Number of atoms $n$ is sampled implicitly via a stop token (p(stop | …) in Eq 7). We will state this in line 245.
>
> > Would it be more appropriate to avoid imposed atom orderings?
>
> We agree, although “all models are wrong, but some are useful.” Autoregressive models with stop tokens can naturally handle variable numbers of atoms, unlike pure diffusion models. We highlight the strong precedent of applying sequential and non-permutation invariant architectures to unordered data across AI, often achieving SOTA results [9,10,11]. While one can train sequential models on all orderings, marginalizing out orderings is intractable for even 20 atoms.
>
> [9] An Image is Worth 16x16 Words: Transformers for Image Recognition at Scale. ICLR 2021
>
> [10] Scaling Autoregressive Models for Content-Rich Text-to-Image Generation. TMLR 2022
>
> [11] Scalable Diffusion for Materials Generation. ICLR 2024
>
> > How were features for space groups and Wyckoff positions derived?
>
> We will add the following to our appendix.
>
> For the space groups, we used one-hot features indicating the 6 lattice centering types, 6 crystal families, 32 crystallographic point groups, whether the space group is chiral, and whether the space group is centrosymmetric. Since this representation ignores fractional translation symmetries and thus cannot differentiate certain space groups, we counted the max number of collisions between space group embeddings (16), arbitrarily indexed colliding embeddings, and created additional one-hot features to avoid collisions. This yielded `6+6+32+1+1+16=62` features. While the last 16 features are not physically motivated, they give the model the capacity to differentiate space groups and learn their relationships.
>
> To featurize a Wyckoff position, we used our space group features (62); one-hot features for the site symmetry symbol (100), number of degrees of freedom in atom coordinates (4), and Wyckoff multiplicity (17); and 48 Fourier features of Wyckoff shapes’ midpoints and vertices (see Sec 4.1). This yielded `62+100+4+17+48=231` features. To compute Fourier features $\nu \in \mathbb{R}^{48}$, we collected the midpoint and vertices of each Wyckoff shape belonging to the Wyckoff position. We used these points $\\{p_i\in\mathbb{R}^{3\times1}\\}_{i=1}^{n_w}$ to compute the following:
>
> \[
> \begin{aligned}
> f &=2\pi \cdot \operatorname{linspace}(\text{start}=1,\text{end}=32,\text{steps}=8) \in \mathbb{R}^{1 \times 8}\\\\
> \nu_i &=\operatorname{flatten}(p_i f)\in \mathbb{R}^{24}\\\\
> \nu &=\frac{1}{n_w}\sum_{i=1}^{n_w} [\sin(\nu_i),\cos(\nu_i)]\in\mathbb{R}^{48}
> \end{aligned}
> \]
>
> > is 1731 the total number of Wyckoff positions across space groups?
>
> Yes. We will edit line 255 accordingly.
>
> > SymmCD symmetry representations could be an alternative. Was any ablation performed on token design?
>
> We performed the ablation, replacing our space group and Wyckoff features with SymmCD’s. On MP20, our features yielded better validity, Wasserstein distance for density, structural divergence, and validation log-likelihoods of Wyckoff positions. SymmCD features yielded a better UN rate, Wasserstein distance for number of elements, and compositional diversity. Results with standard deviations over 3 training runs are below. We believe the result on Wyckoff log-likelihoods motivates our featurization.
> | |Wyckoff log-likelihood|Structure validity|Composition validity|UN rate|W_\rho|W_{N_{el}}|JSD_{d_{Wyckoff}}|CMD|Structure diversity|Composition diversity|
> |---|---|---|---|---|---|---|---|---|---|---|
> |SGEquiDiff|**-0.3206+/-0.0083**|**99.47+/-0.30**|**85.01+/-0.88**|83.40+/-0.79|**0.5128+/-0.1030**|0.2055+/-0.0067|0.0327+/-0.0078|**0.1968+/-0.0078**|0.9149+/-0.0241|15.58+/-0.06|
> |+SymmCD features|-0.4925+/-0.0030|98.43+/-0.18|81.75+/-0.49|**85.23+/-0.35**|0.7565+/-0.0680|**0.1654+/-0.0232**|0.0229+/-0.0039|0.3470+/-0.0150|0.8960+/-0.0116|**16.06+/-0.08**|
>
> > Is LDiff equivalent to the representation of lattice matrices in DiffCSP++?
>
> Yes. We will edit lines 330-331 for clarity.
>
> > "next best" model is misleading. Given acknowledged variability in SUN rate, lines 334–335 is not well-supported
>
> We will rephrase to “achieved the highest evaluated SUN rate”. We are currently running ~800 more DFT calculations per model to address potential variability.
>
> > Beyond efficiency, are there other advantages to the asymmetric unit? The conventional cell may capture more structural information
>
> Computational efficiency is the main advantage to using the ASU. We used the conventional cell and its periodic boundaries for graph construction (lines 1183-1185) to avoid losing structural information and to symmetrize the model with space group equivariance (Eq 3). Our efficiency gains from using the ASU stem from only computing score matching losses on ASU atoms, only storing ASU atoms in the dataset, and only generating atom types/Wyckoff positions of ASU atoms. We tried excluding fractional coordinates from the representation such that atoms in the same orbit have the same embeddings, removing the need to compute embeddings for atoms outside the ASU. However, performance was better with fractional coordinates.
>
> A possible approach to remove the need to compute “non-ASU” atom embeddings while maintaining performance is to replace periodic sinusoidal embedding functions with space group invariant functions [12]. We defer this to future work.
>
> [12] arXiv:2306.05261 (2023)
>
> **Table 1**
> > What does T represent? What is T in the SGEquiDiff row?
>
> T is the inference-time temperature for all discrete distributions, and T=1.0. We will add these to the table.
>
> > There appears to be an inverse relationship between composition validity and UN rate
>
> We observed the uniqueness rate was nearly 100% for all models. Thus, the relationship occurs because exotic compositions are more likely to be novel but also invalid.
>
> > It is incorrect that SymmCD uses fixed templates
>
> Thank you, we will fix that.
>
> > Please report variance across multiple seeds for SGEquiDiff
>
> We trained SGEquiDiff with 3 random seeds per dataset and will update the tables with average metrics and standard deviations, as below.
> | |Sampling time|Structure valid|Composition valid|UN rate|W_rho|W_{N_{el}}|JSD_G|JSD_{d_{Wyckoff}}|CMD|Structure diversity|Composition diversity|
> |---|---|---|---|---|---|---|---|---|---|---|---|
> |MP20|266+/-40|99.4+/-0.30|85.501+/-0.88|83.4+/-0.79|0.5128+/-0.1030|0.2055+/-0.0067|0.1723+/-0.0041|0.0327+/-0.0078|0.1968+/-0.0078|0.9149+/-0.0241|15.58+/-0.06|
> |MPTS52|678+/-58|97.19+/-0.43|78.21+/-0.18|86.5+/-0.7|1.405+/-0.274|0.4409+/-0.0472|0.3179+/-0.0252|0.1539+/-0.0393|0.4263+/-0.0412|0.8490+/-0.0363|16.21+/-0.3|
>
> > Why is Wasserstein distance for density significantly higher for SGEquiDiff
>
> Good question - we aren’t sure. We checked 1D histograms of unit cell volumes and numbers of atoms per unit cell for generated vs ground truth crystals. We did not see qualitative differences.
>
> Some insight can be drawn from the LDiff ablation, which has a comparable density metric with the best baselines. It's possible that (1) iterative refinement from lattice diffusion improves the density metric (aligned with the result of CDVAE, which doesn't diffuse lattices), (2) the higher density metric is an artifact of discretizing the lattice, and/or (3) choosing the lattice before atom types/Wyckoff positions is harder to learn than the reverse.
>
> Regardless, we don't think the higher density metric is significant in practice since both CDVAE and SGEquiDiff attain better SUN rates (Table 1) and RMSDs (Table 3) than DiffCSP++ despite having higher Wasserstein distances for density.
> > The table caption would benefit from more description
>
> We will add the following to table 1 (similar for table 2): “We report metrics of validity for 10,000 sampled crystals; uniqueness and novelty for 1,000 valid crystals; diversity and goodness-of-fit for valid, unique, and novel (UN) crystals; and stability for 100 UN crystals. Stability is defined as having a DFT-predicted $E_\mathrm{hull}$ < 0.1 eV/atom compared to Materials Project v2024.11.14. Error bars are the standard deviation of three training runs. $T$ represents inference-time temperature of discrete distributions in SGEquiDiff.”
>
> ---
> We hope this addresses your questions. If our responses are helpful, we would be grateful if you would consider raising your score.

---

> > ### Comment · Reviewer_mXGs · 2025-08-02
> > **Response to Authors**
> >
> > Thank you for your detailed answers and for running additional experiments to answer my questions. I have increased my score to recommend acceptance.
> >
> > A few follow-up points:
> > - Regarding the atom orderings, instead of training across multiple, it would be interesting to see the effect of removing positional information from the model.
> > - The LaTeX is not rendered for the question on how "How were features for space groups and Wyckoff positions derived?". Can you correct it? I am also not clear about the values mentioned in "one-hot features for the site symmetry symbol (100), number of degrees of freedom in atom coordinates (4), and Wyckoff multiplicity (17); and 48 Fourier features of Wyckoff shapes’ midpoints and vertices". Can you please clarify these values in the parentheses further?

---

> > > ### Author Response · Authors · 2025-08-04
> > >
> > > Thank you for your reply and for raising your score. Below we respond to your follow-up points.
> > >
> > > > Regarding the atom orderings, instead of training across multiple, it would be interesting to see the effect of removing positional information from the model.
> > > >
> > >
> > > If we understand correctly, you are referring to positional encodings commonly employed in transformer architectures. To clarify, we don’t use a positional encoding in our transformer which we will state more explicitly in the revised paper. We also only train on a single ordering of discrete atom tokens (each consisting of a Wyckoff position and atomic number). The ordering is canonicalized by first ordering atoms alphabetically according to Wyckoff letter; ties are broken by sorting in ascending order according to atomic number (lines 245-246).
> > >
> > > > The LaTeX is not rendered for the question on how "How were features for space groups and Wyckoff positions derived?". Can you correct it?
> > > >
> > >
> > > Sorry about that. It renders for us in Google Chrome, but perhaps it's easier to share the raw LaTeX:
> > >
> > > f = 2\pi \cdot \mathrm{linspace(start=1,end=32,steps=8)} \in \mathbb{R}^{1\times8}
> > >
> > > \nu_i = \mathrm{flatten}(p_i f) \in \mathbb{R}^{24}
> > >
> > > \nu = \frac{1}{n_w} \sum_{i=1}^{n_w} [\sin(\nu_i), \cos(\nu_i)] \in \mathbb{R}^{48}
> > >
> > > > I am also not clear about the values mentioned in "one-hot features for the site symmetry symbol (100), number of degrees of freedom in atom coordinates (4), and Wyckoff multiplicity (17); and 48 Fourier features of Wyckoff shapes’ midpoints and vertices". Can you please clarify these values in the parentheses further?
> > > >
> > >
> > > Apologies for the lack of clarity. Across all space groups, we found 100 unique site symmetry symbols, 4 possible numbers of degrees of freedom (0, 1, 2, and 3), and 17 unique values of Wyckoff multiplicities.

---

### Official Review · Reviewer_JT9f · 2025-06-30

**Clarity:** 3
**Significance:** 2
**Originality:** 3
**Rating:** 4
**Confidence:** 3

**Summary:**

This paper introduces SGEquiDiff, a Space Group Equivariant Diffusion model for crystal structure generation. The method enforces space group constraints and provides explicit space group–invariant likelihoods. Experiments on standard crystal datasets demonstrate solid performance.

**Questions:**

I appreciate the motivation and design of the proposed method and believe I understand the core ideas. Given that the method explicitly enforces symmetry constraints, I would expect it to outperform approaches that do not incorporate such inductive biases.

However, I find the experimental section lacking in several key areas: the dataset diversity is limited, comparisons with strong baselines are insufficient, and the model's potential for Crystal Structure Prediction—an important application—is not clearly demonstrated. Moreover, the performance gains are only evident in the S.U.N. metric, while no clear advantage is shown in other evaluation metrics. Strengthening these aspects would significantly enhance the credibility and impact of the work.

## Questions for the authors:

* **Q1** : Why does SGEquiDiff achieve better performance on the S.U.N. metric compared to baselines, yet underperform on most other evaluation metrics? How should the importance of the S.U.N. metric be interpreted relative to the others?

* **Q2** : Can the proposed method be applied to Crystal Structure Prediction? If so, how does it perform compared to existing baselines on this task?

**Ethical Concerns:**

["NO or VERY MINOR ethics concerns only"]

**Final Justification:**

Most of my concerns have been addressed, and the clarifications help. I raise my score to 4.

**Limitations:**

yes

**Paper Formatting Concerns:**

I do not have any concerns regarding the format.

**Quality:**

2

**Strengths And Weaknesses:**

## Strengths
* **S1** : The proposed model allows hard constraints to be incorporated through a space group–invariant likelihood, without requiring local structure relaxation.
* **S2** : The paper presents a carefully constructed and comprehensive review of related work, clearly situating the proposed approach within the existing literature.
* **S3** : The proposed method leverages the domain’s symmetry as an inductive bias. Given this, the improvement in S.U.N. rate is expected and not surprising.


## Weaknesses

I have several concerns regarding the experimental evaluation:

* **W1** : The experiments are conducted on only two datasets, whereas related work such as [1] and [2] typically report results on at least three datasets (e.g., Perov-5, MP20, and Carbon-24). One of the selected datasets—though challenging—lacks thorough baseline comparisons. While the authors note that previous models have not reported results on this dataset, I would expect them to run those baselines using publicly available code to enable a fairer and more comprehensive comparison. Relying on results from just one baseline weakens the experimental support for the proposed method.
* **W2** : Although the proposed method achieves the best performance on the S.U.N. metric, it underperforms on most of the other evaluation metrics compared to baselines.
* **W3** : It is unclear how well the proposed model performs on the Crystal Structure Prediction (CSP) task—a central objective in crystal generation—unlike other models such as [1], [2], and [3], which explicitly evaluate this aspect.


[1] Space Group Constrained Crystal Generation

[2] Crystal Structure Prediction by Joint Equivariant Diffusion

[3] FlowMM: Generating Materials with Riemannian Flow Matching

---

> ### Author Rebuttal · Authors · 2025-07-31
>
> Thank you for your helpful feedback and questions. Below we respond to your questions and concerns.
>
> > **W1** : The experiments are conducted on only two datasets, whereas related work such as [1] and [2] typically report results on at least three datasets (e.g., Perov-5, MP20, and Carbon-24).
> >
>
> We note that Perov-5 and Carbon-24 are not used in more recent works [4,5,6,7,8] (including a followup work to [3]) because they are toy datasets containing unstable crystals; all crystals in Perov-5 have the same fractional coordinates and all crystals in Carbon-24 have only 1 element. In contrast, crystals in MP20 and MPTS52 have all been experimentally observed and span 89 elements of the periodic table with a large diversity of structures. These datasets are thus much more challenging and relevant for materials discovery, becoming the de facto benchmarks for crystal generative models.
>
> [4] A generative model for inorganic materials design. *Nature* **639**, 624–632 (2025)
>
> [5] SymmCD: Symmetry-Preserving Crystal Generation with Diffusion Models (ICLR 2025)
>
> [6] All-atom Diffusion Transformers: Unified generative modelling of molecules and materials (ICML 2025)
>
> [7] Wyckoff Transformer: Generation of Symmetric Crystals (ICML 2025)
>
> [8] FlowLLM: Flow Matching for Material Generation with Large Language Models as Base Distributions (NeurIPS 2024)
>
> > One of the selected datasets—though challenging—lacks thorough baseline comparisons. While the authors note that previous models have not reported results on this dataset, I would expect them to run those baselines using publicly available code to enable a fairer and more comprehensive comparison. Relying on results from just one baseline weakens the experimental support for the proposed method.
> >
>
> We agree. To strengthen our results, we modified the codes of DiffCSP and DiffCSP++ and benchmarked them on MPTS-52, using the same diffusion corrector step size as for MP20 and dropping the default batch size of DiffCSP++ to 96 to avoid OOMs. We additionally ran SymmCD on our own hardware. We failed to train CDVAE on MPTS-52 due to errors during graph construction arising from noised atoms becoming isolated within the model’s cutoff radius. As an additional diversity metric, we followed SymmCD and report the fraction of unique and novel templates with respect to the training data (where a template is defined as a space group and a multiset of occupied Wyckoff positions) out of 1000 structurally and compositionally valid sampled crystals. We found that SGEquiDiff had a significantly higher unique and novel template rate than baselines while maintaining high structural validity. Although DiffCSP++ yielded slightly higher structural validity, the model uses ground truth templates from training data which explains its extremely low rate of novel templates. The updated table 2 is below:
>
> |  | Sampling time | Structural validity | Composition validity | U.N. rate | W_\rho | W_{N_{el}} | JSD_G | JSD_{d_{Wyckoff}} | CMD | Structure diversity | Composition diversity | Unique and novel template rate |
> | --- | --- | --- | --- | --- | --- | --- | --- | --- | --- | --- | --- | --- |
> | DiffCSP | 467 | 67.47 | 55.8 | 79.8 | 1.189 | 0.5006 | 0.6900 | **0.0818** | 0.4836 | **0.8621** | **16.48** | 0.198 |
> | DiffCSP++ | 1230 | **99.87** | 77.52 | 86.9 | **0.8244** | 0.3692 | **0.2759** | 0.1154 | **0.3812** | **0.8457** | 15.69 | 0.010 |
> | SymmCD | **210** | 87.11 | **78.18** | **90.2** | 1.126 | **0.3506** | **0.2730** | 0.1146 | 0.4775 | 0.7843 | 15.36 | 0.148 |
> | SGEquiDiff | 678 +/- 58 | 97.19 +/- 0.43 | **78.21 +/- 0.18** | 86.5 +/- 0.7 | 1.405 +/- 0.274 | 0.4409 +/- 0.0472 | 0.3179 +/- 0.0252 | 0.1539 +/- 0.0393 | 0.4263 +/- 0.0412 | **0.8490 +/- 0.0363** | **16.21 +/- 0.32** | **0.333 +/- 0.021** |
>
> > **Q1** : Why does SGEquiDiff achieve better performance on the S.U.N. metric compared to baselines, yet underperform on most other evaluation metrics? How should the importance of the S.U.N. metric be interpreted relative to the others?
> >
> We believe that our inclusion of a comprehensive set of metrics (including new ones) is a strength of our work. We also respectfully disagree that SGEquiDiff underperforms on most other evaluation metrics - we believe its performance is on par with other SOTA models that we benchmarked. None of the baseline models achieved SOTA performance on more than 4 of the 12 metrics (although we don’t believe that disqualifies them from being useful and interesting). SGEquiDiff achieved SOTA (or comparable) performance on 4: Wasserstein distance for number of elements, Jensen-Shannon divergence to space groups and Wyckoff dimensions, and the S.U.N. metric. On several other metrics, SGEquiDiff’s performance is competitive in any practical sense (e.g., structure validity, composition validity, composition diversity). We highlight that SGEquiDiff (5.5M parameters) is the second smallest model (CDVAE has 4.9M, DiffCSP(++) has 12.3M, SymmCD has 60.4M, and Mattergen has 44.6M), which we will add to the paper.
>
> Ultimately, the most important metric measures the ability to produce novel and synthesizable crystals. For this reason, the S.U.N. rate [4] is the most important metric as it leverages quantum chemistry calculations to assess stability.
> > **W3** : It is unclear how well the proposed model performs on the Crystal Structure Prediction (CSP) task—a central objective in crystal generation—unlike other models such as [1], [2], and [3], which explicitly evaluate this aspect.
> **Q2** : Can the proposed method be applied to Crystal Structure Prediction? If so, how does it perform compared to existing baselines on this task?
> >
>
> CSP was not a focus of our work because we believe the current formulation of the task in ML papers is slightly misaligned with real world application. DiffCSP [2], FlowMM [3], and other models assume that the number of atoms per unit cell will be known beforehand, which is not true in practice. For example, iron exists in BCC (2 atoms per conventional cell), FCC (4 atoms per conventional cell), and at high pressures, HCP (6 atoms per conventional cell). DiffCSP++ [1] relies on training examples of templates which specify number of atoms, Wyckoff positions, and atom types; thus, performance will worsen in composition spaces with sparse training data. However, these sparse spaces are exactly where experimental materials discovery campaigns are aimed.
>
> Nevertheless, we agree that CSP is an important problem for materials discovery and have done a preliminary evaluation of SGEquiDiff in the same way as prior crystal generative model papers, calculating match rate and RMSE based on a single generated crystal per composition. Due to the rebuttal time constraints, we only evaluated the CSP task on MP20 using the pre-trained SGEquiDiff+LDiff model from Table 1 with DiffCSP++’s CSPML method for template selection. Without any CSP-specific training or hyperparameter tuning (unlike the baselines), we observed competitive performance (see table below). We would be happy to revisit this experiment, performing full tuning to ensure representative and meaningful results for the revised paper.
>
> We note that SGEquiDiff could be alternatively adapted for CSP by reordering the factorization in Eq 5 or by supplying composition as conditioning during training. Learning composition (rather than strictly enforcing it) may be desirable since an arbitrary composition is not guaranteed to host a stable crystal (accordingly, experimentally observed compositions can be averages of multiple phases).
>
> |  | MR (%) | RMSE |
> | --- | --- | --- |
> | CDVAE | 33.90 | 0.1045 |
> | DiffCSP | 51.49 | 0.0631 |
> | FlowMM | 61.39 | 0.0566 |
> | DiffCSP++ (+CSPML) | 70.58 | 0.0272 |
> | SGEquiDiff (+LDiff,CSPML) | 68.52 | 0.0459 |
>
> ---
> We hope this addresses your concerns. If our responses are helpful, we would be grateful if you would consider raising your score.

---

> > ### Comment · Reviewer_JT9f · 2025-08-03
> > **Response to Reviewers' rebuttal**
> >
> > Thanks for the detailed response. Most of my concerns have been addressed, and your clarifications help.
> > I will raise my score.

---

> > > ### Author Response · Authors · 2025-08-07
> > >
> > > Thanks for your response and for agreeing to raise your score. As the discussion period closes, we would like to send a friendly reminder to complete the mandatory acknowledgement and score update. Feel free to let us know if there are any remaining questions or concerns we can address.

---

### Official Review · Reviewer_56AU · 2025-07-01

**Clarity:** 2
**Significance:** 3
**Originality:** 3
**Rating:** 5
**Confidence:** 4

**Summary:**

The paper proposes a new generative model for de-novo crystalline materials generation by modelling the most asymmetric unit of the conventional unit cell, rather than the primitive unit cell as done in most recent generative models. The generative process is split in an autoregressive way: first, they compute the Wyckoff "description" of the crystal (space group, Wyckoff position, lattice parameters) and atom type and then they run diffusion on the fractional coordinates (similar to crystal-structure prediction). In particular, they introduce the space-group wrapped normal distribution which is used to train the score network, which ensures that the resulting vector field is equivariant to the crystal’s space group symmetries. Results on MP-20 shows that they are able to improve S.U.N rate metric compared to Mattergen computed using DFT on 100 samples.

**Questions:**

1. I am bit confused about eq. 10 and eq. 11 in appendix. The two equations means that the function $f(x)$ is at the same time invariant and equivariant to $R$. But then this should hold just for trivial $R$, like if $R$ is the identity, right? Am I missing something here? Or is it just because we are considering group actions from the group stabilizer (which leaves the input unchanged and therefore the output is fixed) that this holds? Clarifying this would help avoid confusion.
2. I'm curious about the decision to learn $p(G)$ (the distribution over the space groups), rather than setting it to the empirical distribution of the training set. Doesn't the optimization of the logits lead to the empirical distribution?
3. I think it will be interesting and useful for the community to understand where the most gains in performance comes from. Although there are some differences in the input used across all the different generative models, many of them uses the same EGNN backbone architecture (from Satorras et al.). Is the FaeNet specifically needed to enforce some of the constraints you want to impose? Have you try to compare the performance using the same backbone architecture as the other baselines, to isolate the benefit of the different architecture?
4. Space groups define specific constraints to the lattice angles. How do you enforce those constraints during sampling? Are they hard-coded?
5. Is the featurization of the space group different from the one used in SymmCD? If so, I think it would be helpful to explicitly state that and highlight differences.
6. The EquiCSP paper highlights that the score used by DiffCSP is not periodic translation invariant, meaning that the score for a position $x$ and a periodic translation of it are different. Does your score formulation address this issue or does it suffer still from this mismatch?

Minor:

1. Missing the log in the score matching loss $L_X$ and also in line 280

I am willing to increase my score if my doubts are solved.

**Ethical Concerns:**

["NO or VERY MINOR ethics concerns only"]

**Final Justification:**

After going through the author's rebuttals and the answers they have provided to the other reviewers, I feel like they solved all my doubts. I think that the additional ablations and the rewriting they have promised will make the paper more informative and will help reproducibility.

**Limitations:**

yes

**Quality:**

3

**Strengths And Weaknesses:**

1. The idea of modelling the most asymmetric unit cell and the symmetry transformations (given by space group and Wyckoff positions) is an exciting research direction. Given the many symmetries constraints inherent in crystalline materials, restricting fractional coordinates to respect these constraints rather than modeling them over a continuum makes a lot of sense. In this context, the autoregressive approach considered in the paper is a novel contribution. Moreover, the ability to generate a Wyckoff description that rigorously obeys symmetry rules is a particularly strong result.

2. The paper can benefit from more clarity as some parts were difficult to fully understand.Providing more details about the training procedures of the four different components of the model would likely help. I will highlight some observations here and add some additional question in the Question section
	- the notation used for functions and group actions is sometimes overlapping, which can be misleading at times
	- Line 185 to 188: it's not completely clear what the message is there. Is the main point just that space group equivariant vector field does not make any transition between different Wyckoff position dimensions (meaning that if the Wyckoff position is 1D the vector field will stay on that line), whereas the usual SDE approach does not respect this due to the Brownian motion? The use of the term *motivating* feels somewhat strange in this context.
	- Line 237 to 240: it's not completely clear how the model used to sample the lattice parameters is trained. Once one lattice parameter is generated, are you using a noisy version of it as conditioning to generate the others? Clarifying that process would be helpful
	- The table captions could be expanded to be more self-contained, for example by being be more specific about what is presented in them.

---

> ### Author Rebuttal · Authors · 2025-07-31
>
> Thank you for the helpful feedback and questions. Below we aim to answer questions and address your concerns. Please let us know if we can clarify anything further.
>
> > The notation used for functions and group actions is sometimes overlapping, which can be misleading at times
>
> Thank you for pointing this out. If we understand correctly, you are referring to the usage of $g(t)$ in equations 1 and 2 overlapping with the usage of $g$ as a group element. To avoid confusion, we will replace $g(t)$ in equations 1 and 2 with $\omega (t)$.
>
> > Line 185 to 188: it's not completely clear what the message is there.
>
> We agree this is a confusing point and will update the text for clarity. The intended message is that, for example, if a 0D Wyckoff position (a point) lies in a 1D Wyckoff position (a line), then a space group equivariant vector field cannot move an atom along the line through the point. This is problematic because we observed examples of space groups where traversing between symmetrically unique regions of a 1D Wyckoff position requires moving through a 0D Wyckoff position. In contrast to, e.g., ODE-based sampling, the use of Gaussian noise (projected onto the tangent space of the Wyckoff position) in SDE-based sampling allows models to stochastically overcome these barriers while maintaining that the marginal distribution over atom coordinates is still space group invariant.
>
> > Line 237 to 240: it's not completely clear how the model used to sample the lattice parameters is trained. Once one lattice parameter is generated, are you using a noisy version of it as conditioning to generate the others? Clarifying that process would be helpful
>
> During training, we maximize the log probability of the next ground truth lattice parameter conditioned on noisy versions of the previous ground truth lattice parameters (where noise is masked such that space group constraints are respected). At sampling time, noising is not applied. We will add this to the revised paper.
>
> > The table captions could be expanded to be more self-contained, for example by being be more specific about what is presented in them.
>
> We agree and will append the following text to the caption of table 1: “We report metrics of validity for 10,000 sampled crystals; uniqueness and novelty for 1,000 valid crystals; diversity and goodness-of-fit distributional distances for valid, unique, and novel crystals; and stability for 100 valid, unique, and novel crystals. Stability is defined as having a DFT-predicted $E_\mathrm{hull}$ < 0.1 eV/atom with respect to the Materials Project v2024.11.14. Error bars are the standard deviation of three training runs. $T$ represents the inference-time temperature of discrete distributions in SGEquiDiff.” We will append similar text to the caption of table 2.
>
> > 1. I am bit confused about eq. 10 and eq. 11 in appendix. The two equations means that the function $f(x)$ is at the same time invariant and equivariant to $R$. But then this should hold just for trivial $R$, like if $R$ is the identity, right? Am I missing something here? Or is it just because we are considering group actions from the group stabilizer (which leaves the input unchanged and therefore the output is fixed) that this holds? Clarifying this would help avoid confusion.
>
> Yes, as you mentioned, equations 10 and 11 hold because as in line 1133, we are only considering group actions from the stabilizer group. To avoid confusion, we will update the notation to read $R_{g_x}$ and $v_{g_x}$ instead of $R$ and $v$.
>
> > 2. I'm curious about the decision to learn $p(G)$ (the distribution over the space groups), rather than setting it to the empirical distribution of the training set. Doesn't the optimization of the logits lead to the empirical distribution?
>
> We agree (although we applied early stopping based on validation performance as a soft regularizer). For simplicity, we have changed our code to sample space groups directly from the training distribution. We observed no changes to our experimental results.
>
> > 3. I think it will be interesting and useful for the community to understand where the most gains in performance comes from. Although there are some differences in the input used across all the different generative models, many of them uses the same EGNN backbone architecture (from Satorras et al.). Is the FaeNet specifically needed to enforce some of the constraints you want to impose? Have you try to compare the performance using the same backbone architecture as the other baselines, to isolate the benefit of the different architecture?
>
> Thank you for the question and suggestion. FAENet is not required to enforce space group equivariance. From lines 174-175, any periodic translation invariant architecture suffices, including the EGNN backbone used in DiffCSP and others. We have conducted the ablation on MP20 as suggested using the aforementioned EGNN backbone (”CSPNet”). For fair comparison, we held constant the number of message passing steps (5), the number of Gaussian edge frequencies (96), and the number of trainable parameters (2.2M) by reducing CSPNet’s hidden dimension from 512 to 224. We observed our architecture performed slightly better than CSPNet on the structural validity and U.N. metrics (possibly due to our use of Cartesian distances as edge features) while performing the same on other structural metrics. We will include the results (below) in our appendix, where standard deviations over 3 training runs are reported and best result is bold.
>
> |  | Structural validity | U.N. rate | CMD | Structure diversity |
> | --- | --- | --- | --- | --- |
> | SGEquiDiff | **99.47 +/- 0.30** | **83.40 +/- 0.79** | 0.1968 +/- 0.0078 | 0.9149 +/- 0.0241 |
> | +CSPNet | 98.84 +/- 0.12 | 81.33 +/- 1.04 | 0.2052 +/- 0.0014 | 0.9286 +/- 0.0259 |
>
> > 4. Space groups define specific constraints to the lattice angles. How do you enforce those constraints during sampling? Are they hard-coded?
>
> Constraints are enforced in the same way during training and sampling with hard-coded masking and addition of any pre-determined angles. We will update lines 219-220 to reflect this.
>
> > 5. Is the featurization of the space group different from the one used in SymmCD? If so, I think it would be helpful to explicitly state that and highlight differences.
>
> Yes, our space group featurization is different. SymmCD used binary features indicating the 26 space group symmetry operations, the 15 symmetry axes, and the 7 lattice systems. This yielded `7 + 26 x 15 = 397` features. We used more abstracted one-hot features indicating the 6 lattice centering types, 6 crystal families, 32 crystallographic point groups, whether the space group is chiral, and whether the space group is centrosymmetric. Since this representation ignores fractional translations from glide and screw symmetries, we counted the maximum number of collisions between space group representations (16), arbitrarily indexed colliding space groups, and created additional one-hot features to avoid collisions. This yielded `6 + 6 + 32 + 1 + 1 + 16 = 62` features. While the last 16 features are not physically motivated, they give the model the capacity to differentiate space groups and learn their relationships.
>
> As an ablation, we replaced our space group and Wyckoff position features with those used in SymmCD. On MP20, we found our featurization yielded better structure validity, composition validity, Wasserstein distance for atomic density, structural divergence, and validation log-likelihoods of ground truth Wyckoff positions. The SymmCD features yielded a slightly better U.N. rate, Wasserstein distance for number of elements, and compositional diversity. The results (with standard deviations over 3 training runs) are summarized below where best result is bold. We believe the particularly strong result on Wyckoff log-likelihoods motivates our featurization.
> |  | Validation Wyckoff log-likelihood | Structural validity | Composition validity | U.N. rate | W_\rho | W_{N_{el}} | JSD_{d_{Wyckoff}} | CMD | Structure diversity | Composition diversity |
> | --- | --- | --- | --- | --- | --- | --- | --- | --- | --- | --- |
> | SGEquiDiff | **-0.3206 +/- 0.0083** | **99.47 +/- 0.30** | **85.01 +/- 0.88** | 83.40 +/- 0.79 | **0.5128 +/- 0.1030** | 0.2055 +/- 0.0067 | 0.0327 +/- 0.0078 | **0.1968 +/- 0.0078** | 0.9149 +/- 0.0241 | 15.58 +/- 0.06 |
> | +SymmCD features | -0.4925 +/- 0.0030 | 98.43 +/- 0.18 | 81.75 +/- 0.49 | **85.23 +/- 0.35** | 0.7565 +/- 0.0680 | **0.1654 +/- 0.0232** | 0.0229 +/- 0.0039 | 0.3470 +/- 0.0150 | 0.8960 +/- 0.0116 | **16.06 +/- 0.08** |
>
> > 6. The EquiCSP paper highlights that the score used by DiffCSP is not periodic translation invariant, meaning that the score for a position and a periodic translation of it are different. Does your score formulation address this issue or does it suffer still from this mismatch?
>
> Yes, our score is periodic translation invariant. Following our paper’s notation, the space group wrapped normal (SGWN) distribution and its score can be written as
>
> \[
> \begin{aligned}
> p_\mathrm{SGWN}(x_t|x_0) &\propto \sum_{t_L \in \mathbb{Z}^3} \sum_{\\{R_j, t_j\\} \in G/T_L} \exp\left(\frac{-\lVert x_t - (R_j x_0 + t_j + t_L) \rVert_F^2}{2\sigma_t^2} \right) \\\\
> \nabla_{x_t}\log p_\mathrm{SGWN}(x_t|x_0) &= \frac{\sum_{t_L \in \mathbb{Z}^3} \sum_{\\{R_j, t_j\\} \in G/T_L} \exp\left(\frac{-\lVert x_t - (R_j x_0 + t_j + t_L) \rVert_F^2}{2\sigma_t^2} \right) (R_jx_0 + t_j + t_L - x_t)}{\sigma_t^2 p_\mathrm{SGWN}(x_t|x_0)}.
> \end{aligned}
> \]
>
> Applying a lattice translation $t^\prime_L \in \mathbb{Z}^3$ as $x_t \rightarrow x_t + t^\prime_L$ will leave the SGWN and its score invariant since $t^\prime_L$ can be absorbed by the infinite sum $\sum_{t_L \in \mathbb{Z}^3}$.
>
> > Missing the log in the score matching loss and also in line 280
>
> Thank you for catching this!
>
> ---
> We hope this addresses your questions. If our responses are helpful, we would be grateful if you would consider raising your score.

---

> > ### Comment · Reviewer_56AU · 2025-08-01
> >
> > I want to thank the authors for their thorough and detailed responses to the points I raised in my review. I also want to thank them for running the additional ablations. I will raise my score accordingly. I just have the following follow up comments:
> >
> > - **About notation** I apologize for not being more specific in my initial review. I feel like the confusion arises from the notation used for the group action: it is defined as $g(\cdot)$ and sometimes it appears as $g(x)$ and other times as a multiplication $gx$. I don't think it is a problem having the diffusion term in the SDE denoted by $g$. since I seem to be the only one noting this, it might not be a high-priority issue.
> >
> > - **Clarification of lines 185 to 188** Am I understanding correctly that if, during the generation of an atom constrained to a 1D Wyckoff position (a line) then if the position moves to a 0D Wyckoff position (a point) then an equivariant space group vector field cannot move the atom away from that point? In Figure 3 it's when the atom  atom reaches an endpoint of the line. the projected Gaussian noise introduced by the SDE helps to "unstick" or perturb the atom away from the 0D Wyckoff position, right? On a related note, I have another possibly naive question: when referring to the tangent space, I mostly think of it in the context of differential/Riemannian geometry, but in your case you are primarily referring to the subspace defined by the Wyckoff position (point, line, etc), right?

---

> > > ### Author Response · Authors · 2025-08-04
> > >
> > > Thank you for your prompt response and for raising your score. Below we respond to your remaining comments:
> > >
> > > > **About notation** I apologize for not being more specific in my initial review. I feel like the confusion arises from the notation used for the group action: it is defined as $g(\cdot)$ and sometimes it appears as $g(x)$ and other times as a multiplication $gx$. I don't think it is a problem having the diffusion term in the SDE denoted by $g$. since I seem to be the only one noting this, it might not be a high-priority issue.
> > > >
> > >
> > > Thanks for the clarification and for pointing out the inconsistent notation. We will revise the text to consistently use one of the two notations.
> > >
> > > > **Clarification of lines 185 to 188** Am I understanding correctly that if, during the generation of an atom constrained to a 1D Wyckoff position (a line) then if the position moves to a 0D Wyckoff position (a point) then an equivariant space group vector field cannot move the atom away from that point? In Figure 3 it's when the atom atom reaches an endpoint of the line. the projected Gaussian noise introduced by the SDE helps to "unstick" or perturb the atom away from the 0D Wyckoff position, right?
> > > >
> > >
> > > That is exactly right.
> > >
> > > > On a related note, I have another possibly naive question: when referring to the tangent space, I mostly think of it in the context of differential/Riemannian geometry, but in your case you are primarily referring to the subspace defined by the Wyckoff position (point, line, etc), right?
> > > >
> > >
> > > Yes that is correct - in this case, we have 3D Euclidean space as a (trivial) 3-manifold, a plane as a 2-manifold, a line as a 1-manifold, or a point as a 0-manifold (whose tangent space is just $\\{\mathbf{0}\\}$). These are simple examples of manifolds since they are flat and not just locally Euclidean (as required to be a manifold) but globally so (ignoring artificial boundaries of the unit cell or asymmetric unit which we can always wrap back to). We refer to tangent spaces in the same way as they are used in differential geometry.

---

### Official Review · Reviewer_YxwH · 2025-07-02

**Clarity:** 4
**Significance:** 3
**Originality:** 4
**Rating:** 5
**Confidence:** 4

**Summary:**

This work proposes a diffusion model for crystals with explicit space-group equivariance. The authors do so by using a key-fact that space-group wrapped normal (SGWN) distributions are invariant with respect. A factorization of the generative distribution is proposed. Important physical constraints are enforced. For example, lattice cell volume positivity is imposed by updating the support of the conditional generative distribution for the lattice cell parameter $\gamma$. An empirical evaluation on MP20 is carried out, and comparisons with other crystal generative models are made. A metric is proposed, involving the proportion of stable, unique, and novel (SUN) structures generated by the model. It is show that the proposed model improves upon others using this metric.

**Questions:**

See weaknesses. I am willing to place greater confidence in these findings if the number of structure used for the SUN metric was increased, e.g. by another order of magnitude.

**Ethical Concerns:**

["NO or VERY MINOR ethics concerns only"]

**Final Justification:**

The author's adequately addressed my concerns in the rebuttal, and are working to add more DFT calculations in the next revision.

**Limitations:**

I thank the authors for discussing the weaknesses/limitations of their work in some parts of their paper.

**Quality:**

4

**Strengths And Weaknesses:**

**Strengths**

Space-groups are an important topic in the theory of crystals. Equivariance is an important topic in modern machine-learning. This work addresses the intersection of these two topics in a timely manner.

Good concise intro on the immense size of crystal space. Good motivation with Neumann's principle, a classic reference in materials science.

**Weaknesses**

As noted by the authors, one can relax the outputs of non-"space group equivariant" generative models to get high-symmetry structures. My interpretation of this is that the proposed modelling approach is not strictly necessary to achieve their stated outcomes.

The empirical advantage is not so clear. Only $N=100$ DFT relaxations were used in assessing their SUN metric. Therefore I have some doubts about the strength of the findings.


**Quality**

Overall the quality of this paper is high. I have confidence that the technical aspects of the paper are correct.


**Clarity**

The writing is clear. Theoretical claims appear correct. Claims about improved empirical performance appear supported.




**Significance**

This work could be impactful as it addresses the relevant problem of adding space groups to equivariant generative models for crystal generation.

**Originality**

The work strikes me as original, well-motivated, and well-carried out.

---

> ### Author Rebuttal · Authors · 2025-07-31
>
> Thank you for the thoughtful review and encouraging comments. Below we try to address your concerns.
>
> > As noted by the authors, one can relax the outputs of non-"space group equivariant" generative models to get high-symmetry structures. My interpretation of this is that the proposed modelling approach is not strictly necessary to achieve their stated outcomes.
>
> If we understand correctly, this comment refers to our statement in lines 375-376. To clarify, we believe there are interesting avenues for future work that explore the boundary between space group-constrained models and non-constrained ones to potentially combine the strengths of both. For example, one might train a non-equivariant model with the equivariant scores of our Space Group Wrapped Normal. Furthermore, one might use a non-equivariant model during training, but then symmetrize it with equation 3 to yield space group equivariant scores at inference time. Such approaches could potentially unlock a broader diversity of model architectures able to obtain high symmetry structures. We believe that this capacity to inspire new modeling approaches is a core strength of our work rather than a weakness.
>
> > The empirical advantage is not so clear. Only N=100 DFT relaxations were used in assessing their SUN metric. Therefore I have some doubts about the strength of the findings.
>
> We agree and are currently running several hundred (~800) more DFT calculations per model. Unfortunately we could not obtain the results by the end of the rebuttal period.

---

> > ### Comment · Reviewer_YxwH · 2025-08-04
> > **rebuttal received**
> >
> > I appreciate the author's response to my question about non-constrained models.
> >
> > I encourage the authors to include more DFT calculations in the next revision.
> >
> > I maintain my score.

---

### Decision · Program_Chairs · 2025-09-17

**Decision:**

Accept (poster)

**Comment:**

The paper proposes a new equivariant diffusion method for materials using explicit representation of space group symmetries. The paper's motivation is clear, is well written, and the experimental results show good improvements over prior work. Despite some concerns about evaluations, and comparisons to recent works (like FlowLLM), this paper is a valuable contribution to material generative models. Therefore, I recommend acceptance.